# Differential requirements for mitochondrial electron transport chain components in the adult murine liver

Nicholas P Lesner[1], Xun Wang[1], Zhenkang Chen[1], Anderson Frank[2], Cameron J Menezes[1], Sara House[1], Spencer D Shelton[1], Andrew Lemoff[2], David G McFadden[2,3], Janaka Wansapura[4], Ralph J DeBerardinis[1,3,5,6], Prashant Mishra[1,3,5]*

[1]Children's Medical Center Research Institute, University of Texas Southwestern Medical Center, Dallas, United States; [2]Department of Biochemistry, University of Texas Southwestern Medical Center, Dallas, United States; [3]Harold C. Simmons Comprehensive Cancer Center, University of Texas Southwestern Medical Center, Dallas, United States; [4]Advanced Imaging Research Center, University of Texas Southwestern Medical Center, Dallas, United States; [5]Department of Pediatrics, University of Texas Southwestern Medical Center, Dallas, United States; [6]Howard Hughes Medical Institute, University of Texas Southwestern Medical Center, Dallas, United States

*For correspondence:
prashant.mishra@
utsouthwestern.edu

**Abstract** Mitochondrial electron transport chain (ETC) dysfunction due to mutations in the nuclear or mitochondrial genome is a common cause of metabolic disease in humans and displays striking tissue specificity depending on the affected gene. The mechanisms underlying tissue-specific phenotypes are not understood. Complex I (cI) is classically considered the entry point for electrons into the ETC, and in vitro experiments indicate that cI is required for basal respiration and maintenance of the NAD+/NADH ratio, an indicator of cellular redox status. This finding has largely not been tested in vivo. Here, we report that mitochondrial complex I is dispensable for homeostasis of the adult mouse liver; animals with hepatocyte-specific loss of cI function display no overt phenotypes or signs of liver damage, and maintain liver function, redox and oxygen status. Further analysis of cI-deficient livers did not reveal significant proteomic or metabolic changes, indicating little to no compensation is required in the setting of complex I loss. In contrast, complex IV (cIV) dysfunction in adult hepatocytes results in decreased liver function, impaired oxygen handling, steatosis, and liver damage, accompanied by significant metabolomic and proteomic perturbations. Our results support a model whereby complex I loss is tolerated in the mouse liver because hepatocytes use alternative electron donors to fuel the mitochondrial ETC.

## Editor's evaluation

The commonly accepted view is that complex I of the mitochondrial respiratory chain is required for oxidative phosphorylation. This is an important paper, which discovers that complex I is not required for maintaining metabolically functional mitochondria in the liver, explaining a lack of liver defects in complex I deficiency patients. The study includes compelling data of proteomics and metabolomic tracer analyses representing a resource for a broad community with interest in biochemistry and metabolism of the cell as well as biomedical research. Based on this data the authors propose an alternative source of electrons for the respiratory chain in the mouse liver – the concept of potentially large impact on our understanding of metabolism.

**eLife digest** Mitochondria are specialised structures inside cells that help to convert nutrients into energy. They take electrons from nutrients and use them to power biochemical reactions that supply chemical fuel. Previous studies of cells grown in the laboratory have found that electrons enter this process via a large assembly of proteins in mitochondria called complex I.

Understanding the mechanism of energy production is important, as issues with mitochondria can lead to a variety of metabolic diseases. However, it is still unclear how complex I acts in living animals.

Lesner et al. addressed this knowledge gap by genetically removing a key protein from complex I in the liver of mice. Surprisingly, the animals did not develop any detectable symptoms and maintained healthy liver function. Mice did not seem to compensate by making energy in a different way, suggesting that complex I is not normally used by the mouse liver for this process.

This research suggests that biologists should reconsider the mechanism that mitochondria use to power cells in animals. While the role of Complex I in electron transfer is well established in laboratory-grown cells and some organs, like the brain, it cannot be assumed this applies to the whole body. Understanding energy production in specific organs could help researchers to develop nutrient-based therapies for metabolic diseases.

## Introduction

Mitochondrial ETC disease in humans exhibits a striking tissue specificity, whereby mutations in distinct ETC components are causative for specific and non-overlapping syndromes (*Gorman et al., 2016*). As an example, genetic dysfunction of mitochondrial complex I (cI) is associated with neurological disease including Leigh syndrome (characterized by severe loss of motor abilities, necrotizing brain lesions and early death) and Leber Hereditary Optic Neuropathy (characterized by bilateral loss of vision in early adulthood due to degeneration of the optic nerve) (*Rodenburg, 2016*; *Martín et al., 2005*; *Distelmaier et al., 2009*). The mechanisms underlying sensitivity of the nervous system, with sparing of other organ systems, in the setting of complex I loss is not understood. In contrast, mutations in other mitochondrial ETC components have been associated with multi-organ diseases, and the basis for tissue specificity in those diseases is also unclear (*Ahmed et al., 2018*).

Liver involvement is a common component of mitochondrial diseases, present in approximately 20% of patients (*Lee and Sokol, 2007b*; *Rahman, 2013*), as well as being associated with specific syndromes caused by mutations in distinct ETC-related components (*Valnot et al., 2000*; *Casey et al., 2012*; *Lee and Sokol, 2007a*; *Morris et al., 1998*; *Koh et al., 2018*). Common findings in affected patients include hepatomegaly, steatosis, cholestasis and additional metabolic abnormalities associated with liver failure (*Koh et al., 2018*; *Molleston et al., 2013*; *Cloots et al., 2013*; *Chinnery and DiMauro, 2005*). Genetic analysis of patients has revealed mutations in genes which influence multiple components of the ETC (e.g. *POLG*, *LARS*, *DGUOK*) (*Lee and Sokol, 2007a*). With respect to single-complex diseases, mutations in *SCO1*, an assembly protein for mitochondrial complex IV (cIV), and mutations in *BCS1L*, a chaperone protein for mitochondrial complex III, have been associated with liver failure (*Valnot et al., 2000*; *De Meirleir et al., 2003*; *de Lonlay et al., 2001*). Interestingly, single-complex diseases associated with other components of the ETC (complex I and complex II) do not typically present with liver failure, despite their placement directly upstream of cIV.

The ETC functions to transfer electrons (from the oxidation of nutrients) to $O_2$, as a final electron acceptor; this process is coupled to the formation of an electrochemical gradient across the mitochondrial inner membrane and the production of ATP. Mitochondrial complex I is the largest protein complex of the ETC, and is commonly considered the primary entry point for electrons into the ETC: In cultured cells, cI is required for maintenance of cellular redox status (assessed by the $NAD^+/NADH$ ratio) via its role in NADH oxidation, and subsequent electron transfer to ubiquinone (CoQ; *Sullivan et al., 2015*). Complex I is composed of 14 core subunits, plus an additional 31 accessory subunits (*Sharma et al., 2009*). Published mouse models of complex I deficiency have largely focused on accessory subunits, including deletion of *Ndufs4*, *Ndufs6*, and *Ndufa5*, which result in neurological and cardiac phenotypes associated with partial loss of complex I function (*Kruse et al., 2008*; *Peralta et al., 2014*; *Ke et al., 2012*); no liver phenotypes or metabolic perturbations were reported in these mice (*Yang et al., 2020*). A recent report conditionally deleted *Ndufs3*, a core subunit of cI, in skeletal

muscle, resulting in progressive myopathy associated with significant reduction of cI activity (*Pereira et al., 2020*). Thus, to date, mouse models have supported an essential role for cI in vivo in the central nervous system and muscle.

Here, we report that deletion of the core subunit, *Ndufa9*, results in complete loss of cI function, but is well-tolerated in the adult mouse liver. Further analysis reveals intact liver function and oxygen status in $Ndufa9^{-/-}$ animals, which is not accompanied by significant metabolic alterations or changes in redox status. This is in contrast with hepatocyte-specific deletion of the complex IV assembly factor *Cox10,* which results in altered redox status and impaired liver function. In complex IV-depleted livers, we observe the accumulation of long chain acyl-carnitine species, without significant alterations in other potential electron donors, suggesting that fatty acids are an electron donor to the hepatic ETC. These results support a model in which cI is not an essential upstream component of the ETC in adult mammalian liver, in contrast to the conventional view of cI function.

## Results

### Mitochondrial complex I function is dispensable for homeostasis of adult murine hepatocytes

To understand how liver cells respond to loss of complex I, we made use of a conditional allele of the core subunit *Ndufa9. Ndufa9* deletion in cultured mouse embryonic fibroblasts results in loss of mitochondrial respiratory function, as well as a significant drop in the $NAD^+$/NADH ratio (*Figure 1—figure supplement 1A, B*), indicating that *Ndufa9* is critical for redox status in these cultured cells. We deleted *Ndufa9* in livers of adult mice, making use of adeno-associated virus 8 (AAV8)-mediated Cre recombination driven by the liver specific *Serpina7* promoter in $Ndufa9^{flox/flox}$ animals (see Materials and methods). $Ndufa9^{flox/flox}$ adult animals were injected with control AAV8-GFP virus (hereafter '$Ndufa9^{f/f}$') or AAV8-Cre virus (hereafter '$Ndufa9^{-/-}$'). At 4 weeks post-AAV administration, AAV-Cre injected animals exhibited loss of *Ndufa9* mRNA and protein in hepatocytes (*Figure 1A and B*), as well as loss of complex I activity in mitochondrial lysates (*Figure 1C*). Thus, our experimental model results in severe inhibition of complex I function.

At 4 weeks post-AAV administration, $Ndufa9^{-/-}$ livers displayed no differences in gross physiology (*Figure 1D*) or liver/body weight ratio (*Figure 1F*). Histological analysis of liver sections revealed no significant pathology (H&E), including no evidence of steatosis (Oil Red O staining) or glycogen loss (PAS staining) (*Figure 1E*). Liver function was maintained in $Ndufa9^{-/-}$ livers as indicated by unaltered levels of fasting blood glucose (*Figure 1G*) and circulating indicators of liver function and damage (*Figure 1H, I*). Similar results were observed in mice at 2 weeks post-AAV injection (*Figure 1—figure supplement 2*) and 8 weeks post-AAV injection (*Figure 1—figure supplement 3*). Thus, complex I loss is well-tolerated in adult mouse hepatocytes under homeostatic conditions, with no discernable effects on tissue function, or evidence of tissue damage.

Measurement of steady-state ATP levels did not reveal any changes in complex-I-deficient livers (*Figure 2A*). Complex I inhibition is often associated with elevated reactive oxygen species (*Li et al., 2003*; *Fato et al., 2009*). In $Ndufa9^{-/-}$ livers, we observed a small, but not statistically significant, decrease in the reduced to oxidized glutathione (GSH / GSSG) ratio, indicating that oxidative stress is not significantly induced in the setting of cI loss (*Figure 2B*). Surprisingly, we did not observe altered $NAD^+$/NADH or NADPH/$NADP^+$ ratios in $Ndufa9^{-/-}$ liver (*Figure 2C*). We additionally examined compartment-specific redox status by calculating pyruvate / lactate and acetoacetate / β-hydroxy-butyrate ratios, which are indicative of cytosolic and mitochondrial redox status respectively, neither compartment display evidence of altered redox status (*Figure 2D*). A recent report suggests that circulating α-hydroxybutyrate levels serves as a marker of hepatic redox stress (*Goodman et al., 2020*). $Ndufa9^{-/-}$ animals did not display alterations in tissue or plasma α-hydroxybutyrate levels (*Figure 2E*). Thus, complex I function is not required for maintenance of hepatic redox status, in contrast to its well-established role in cultured cells (*Figure 1—figure supplement 1A, B*; *Sullivan et al., 2015*).

To examine potential compensatory mechanisms that may be enabled in the setting of complex I inhibition, we assessed changes in the proteome and metabolome, comparing $Ndufa9^{f/f}$ and $Ndufa9^{-/-}$ whole livers. At 4 weeks post-AAV, our proteomics analysis detected 3001 proteins with high confidence; however, only 11 proteins were significantly upregulated ($\log_2$(fold change)>1, p-val <0.05), and 23 proteins were significantly downregulated ($\log_2$(fold change) < −1; p-val <0.05)

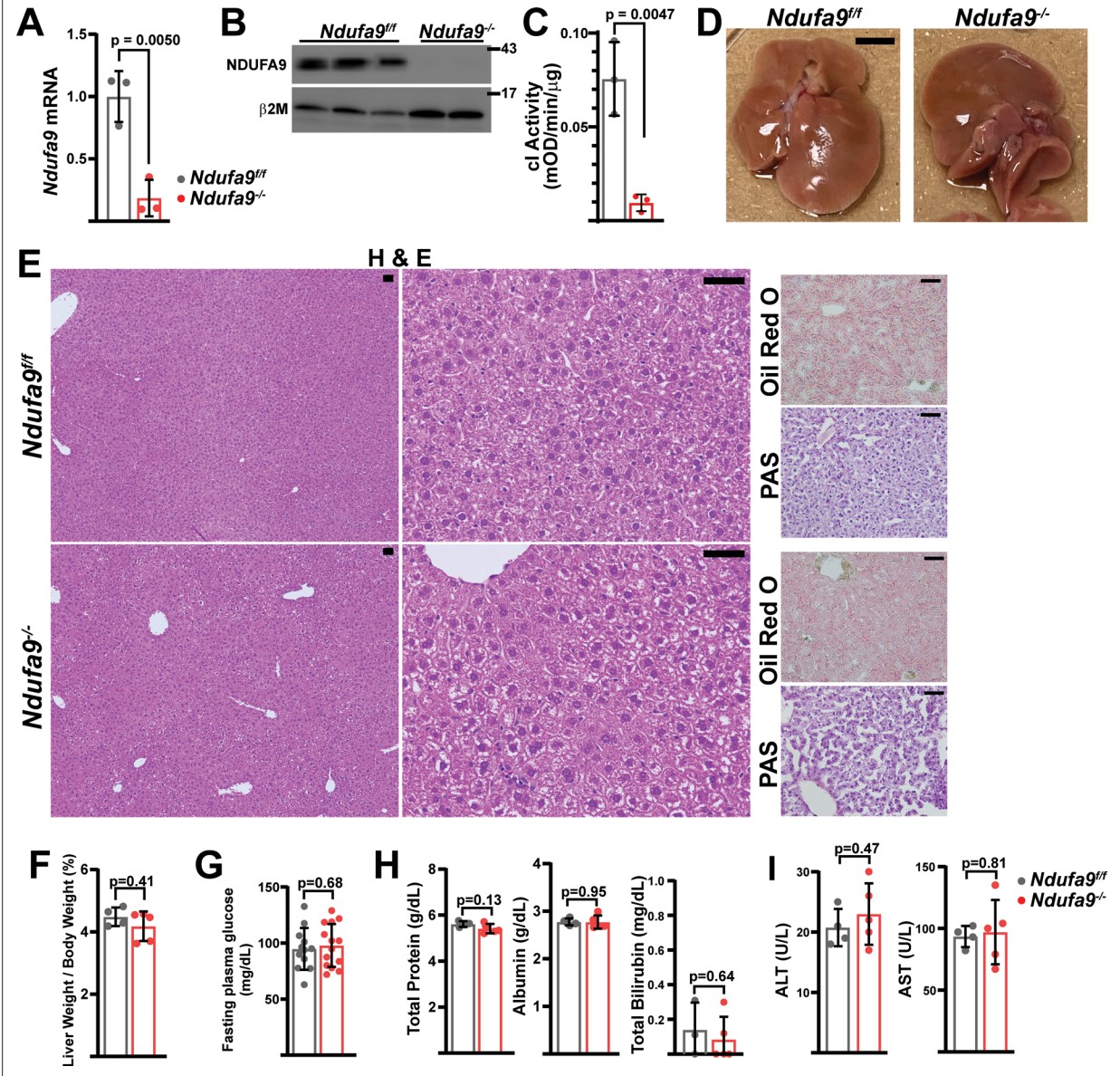

**Figure 1.** Mitochondrial complex I is dispensable in adult murine hepatocytes. (**A**) *Ndufa9* mRNA levels (relative to *β2-microglobulin*; normalized) in isolated hepatocytes of the indicated genotype, assessed by qRT-PCR. n=3 animals per group. The same color scheme is used throughout this figure. (**B**) Ndufa9 protein levels in isolated hepatocytes of the indicated genotype, assessed by Western blot. β2-microglobulin levels are shown as a loading control. MW markers are indicated in kDa. (**C**) Complex I activity measurements from livers of the indicated genotype. n=3 animals per group. (**D**) Representative images of gross liver anatomy in animals of the indicated genotype. Scale bar: 0.5 cm. (**E**) Representative histology (H&E, Oil Red O, PAS staining) images of liver cross sections from animals of the indicated genotype. Scale bar: 50 µm. (**F**) Liver weight (normalized to body weight) from mice of the indicated genotype. n=4–5 animals per group. (**G**) Fasting plasma glucose levels in mice of the indicated genotype. n=12–14 animals per group. (**H**) Circulating plasma markers (total protein, albumin, and total bilirubin) of liver function in mice of the indicated genotype. n=3–5 animals per group. (**I**) Plasma markers (ALT and AST) of liver damage in mice of the indicated genotype. n=4–5 mice per group. All data were collected at 4 weeks post AAV-administration. Statistical significance was assessed using two-tailed t-test (**A,C,G,H,I**) or Mann-Whitney (**F,H**) tests with adjustments for multiple comparisons. All data represent mean +/-standard deviation from biological replicates. Full gel images are provided in *Figure 1—source data 1*. Numerical data for individual panels are provided in *Supplementary file 3*.

The online version of this article includes the following source data and figure supplement(s) for figure 1:

**Source data 1.** Full gel images for *Figure 1*.

**Figure supplement 1.** Metabolic flux analysis of *Ndufa9^f/f^* and *Ndufa9^-/-^* MEFs.

**Figure supplement 2.** Analysis of *Ndufa9^-/-^* livers at 2 weeks post-AAV administration.

**Figure supplement 3.** Analysis of *Ndufa9^-/-^* livers at 8 weeks post-AAV administration.

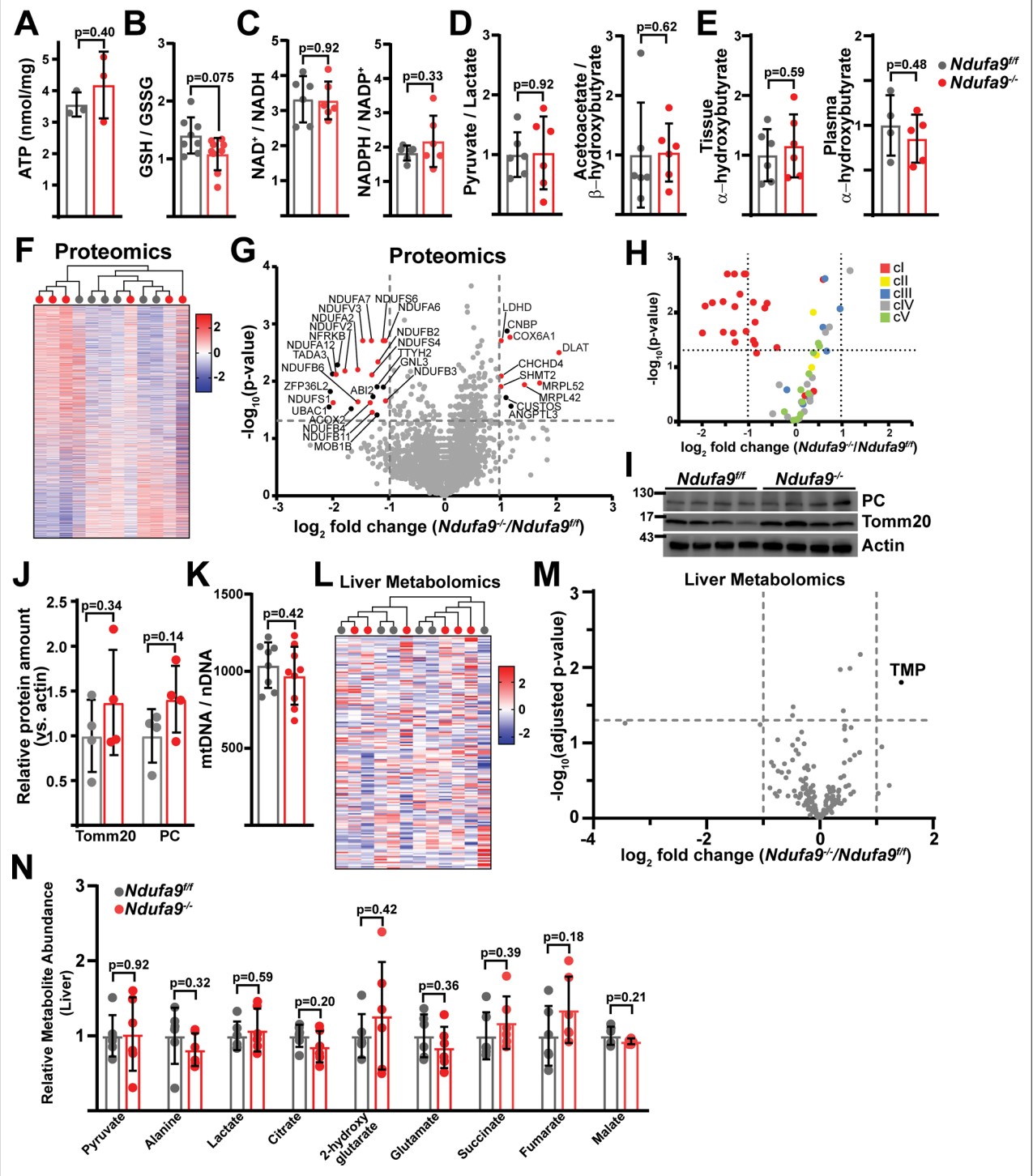

**Figure 2.** Complex I deficiency in adult liver does not impact redox status. (**A**) ATP levels from livers of the indicated genotype. n=3 mice per group. The same color scheme is used throughout this figure. (**B**) Reduced glutathione to oxidized glutathione ratio (GSH/GSSG) in livers of the indicated genotype, as a proxy of oxidative stress. n=8–9 mice per group. (**C**) NAD⁺/NADH and NADPH/NADP⁺ ratios in livers of the indicated genotype. n=6 mice per group. (**D**) Normalized cytosolic (pyruvate / lactate) and mitochondrial (acetoacetate / β−hydroxybutyrate) redox ratios in livers of the indicated genotype. n=6 mice per group. (**E**) Tissue and plasma α−hydroxybutyrate levels in animals of the indicated genotype. n=4–6 mice per group. (**F**) Unsupervised hierarchical clustering of protein abundances in *Ndufa9$^{f/f}$* and *Ndufa9$^{-/-}$* livers, depicted as a heatmap of z-scores. (**G**) Volcano plot of protein abundances changes in *Ndufa9$^{-/-}$* vs. *Ndufa9$^{f/f}$* livers, based on proteomics analysis. Log₂(Fold change) is plotted against the −log₁₀(adjusted p-value) for each protein. Significantly changing proteins (log₂(Fold change)>1 or < −1; Adjusted p-val <0.05) are colored black, and mitochondrial proteins are colored red. n=6 mice per group. (**H**) Volcano plot of abundances for mitochondrial ETC proteins in *Ndufa9$^{-/-}$* vs. *Ndufa9$^{f/f}$* livers, based

*Figure 2 continued on next page*

*Figure 2 continued*

on proteomics analysis. (**I**) Levels of mitochondrial proteins (PC, pyruvate carboxylase, and Tomm20) in livers of the indicated genotype, as assessed by western blot. Actin levels are shown as a loading control. MW markers are indicated in kDa. (**J**) Relative amounts of Tomm20 and PC in livers of the indicated genotype, relative to actin levels. n=4 animals per group. (**K**) Mitochondrial genome (mtDNA) to nuclear genome (nDNA) ratios in livers of the indicated genotype. n=8–9 mice per group. (**L**) Unsupervised hierarchical clustering of liver metabolite abundances in *Ndufa9^f/f* and *Ndufa9^-/-* livers, depicted as a heatmap of z-scores. (**M**) Volcano plot of metabolite abundance changes in *Ndufa9^-/-* vs. *Ndufa9^f/f* livers, based on metabolomics analysis. Log$_2$(Fold change) is potted against the –log$_{10}$(adjusted p-value) for each metabolite. Significantly changing metabolites (log$_2$(Fold change)>1 or < –1; Adjusted p-val <0.05) are colored black. TMP, Thiamine monophosphate. n=6 mice per group. (**N**) Relative abundances of mitochondrial TCA cycle and related metabolites from livers of the indicated genotype. n=6 mice per group. All data were collected at 4 weeks post AAV-administration. Statistical significance was assessed using two-tailed t-test (**A,C,D,E,J,K,N**) or Mann-Whitney (**B**) with adjustments for multiple comparisons. All data represent mean +/-standard deviation from biological replicates. Full gel images are provided in *Figure 2—source data 1*. Numerical data for individual panels are provided in *Supplementary file 1* and *Supplementary file 3*.

The online version of this article includes the following source data and figure supplement(s) for figure 2:

**Source data 1.** Full gel images for *Figure 2*.

**Figure supplement 1.** Analysis of mitochondrial proteomes in complex I deficient livers.

**Figure supplement 2.** Immunofluorescence of hepatic sections.

---

(*Supplementary file 1*). Unsupervised hierarchical clustering was unable to distinguish proteomes from *Ndufa9^f/f* and *Ndufa9^-/-* livers (*Figure 2F*). Of the downregulated proteins, the majority were components of complex I (*Figure 2G and H*; *Supplementary file 1*), consistent with destabilization of cI in the setting of *Ndufa9* subunit loss. We did not observe loss of components from other ETC complexes (*Figure 2H*). Among upregulated proteins, there were a small number of mitochondrial proteins impacting translation (Mrpl52, Mrpl42), electron transport (Cox6a1), and metabolism (Shmt2, Ldhd, Dlat) (*Figure 2G*; *Supplementary file 1*). The small number of abundance changes (outside of complex I components) suggests that the proteome is not drastically changing in response to complex I loss; however, our analysis is only sampling a portion of the total cellular proteome, and it is possible that we are missing some key regulatory proteins in reaction to complex I loss. To further profile the response in *Ndufa9^-/-* livers, we enriched liver lysates for mitochondrial fractions followed by proteomics (*Figure 2—figure supplement 1*). With this method, we were able to detect 738 mitochondrial proteins, which accounts for ~64% of the total mitochondrial proteome (*Supplementary file 1*). We observed significant changes (|log$_2$(fold change)|>1, Adjusted p-val <0.05) in only a small fraction of the mitochondrial proteome (46 proteins;~6% of detected mitochondrial proteins) (*Figure 2—figure supplement 1A*). Complex I subunits tended to be decreased in abundance, and we did not observe significant changes in most subunits of the other complexes (*Figure 2—figure supplement 1B*). We observed statistically significant upregulation of some select components of mitochondrial protein import (TOMM5, TIMM9, TIMM10B) and transcription/translation (MRPS33, MTRES1; *Figure 2—figure supplement 1A*). In addition, we did not detect significant changes in total mitochondrial content in *Ndufa9^-/-* livers, as indicated by measurement of mitochondrial genome and protein content or immunofluorescence (*Figure 2I–K*, *Figure 2—figure supplement 2*).

To assess whether complex I – deficient livers maintained functionality via metabolic compensatory mechanisms, we quantitated the liver metabolome of fasted mice, using LC/Q-TOF mass spectrometry-based metabolomics to measure steady-state levels of >150 tissue (liver) metabolites at 4 weeks post-AAV (*Supplementary file 1*). Unsupervised hierarchical clustering was unable to distinguish metabolomes between *Ndufa9^f/f* and *Ndufa9^-/-* livers (*Figure 2L*), and statistical analysis revealed only a single significant metabolite difference (TMP, thiamine monophosphate; *Figure 2M*, *Supplementary file 1*), and no significant changes in mitochondrial TCA cycle metabolites (*Figure 2N*). The rationale for the observed increase in TMP is currently unclear, but may be related to a reduced need for thiamine pyrophosphate which is a cofactor for NAD(H)-linked dehydrogenases.

In culture, complex I is required for recycling of NADH to NAD$^+$ in order to maintain glycolytic and TCA cycle activity (*Lesner et al., 2020*). We therefore assessed glucose contribution to glycolytic and TCA cycle metabolites via steady-state euglycemic infusions of [U-$^{13}$C]glucose (*Figure 3A and B*). In these experiments, glucose can contribute to TCA cycle metabolites via glycolysis, PDH (pyruvate dehydrogenase), and PC (pyruvate carboxylase) activity; in addition, extrahepatic production of lactate can be metabolized by the liver and contribute to both TCA cycle metabolites and glycolytic intermediates via gluconeogenic pathways (*Figure 3A*). Infusion of $^{13}$C uniformly labeled glucose allows us to

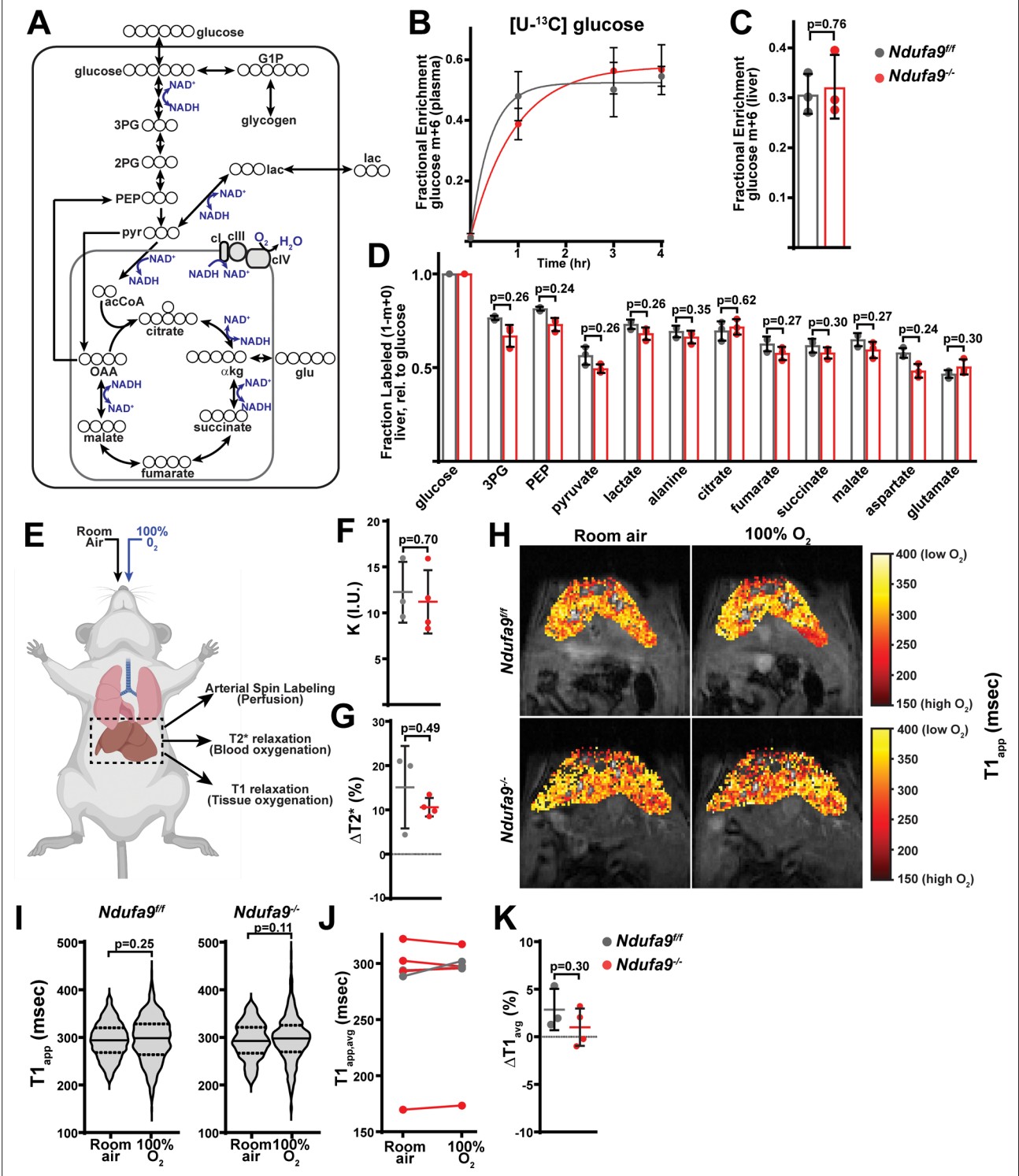

**Figure 3.** Complex I deficiency in adult liver does not impact glucose utilization or oxygen status. (**A**) Schematic of potential fates of glucose and lactate within central carbon metabolism. Carbon atoms are indicated with circles for each compound. G1P, glucose-1-phosphate; 3 PG, 3-phosphoglycerate; 2 PG, 2-phosphoglycerate; PEP, phosphoenolpyruvate; pyr, pyruvate; lac, lactate; acCoA, acetyl-CoA; OAA, oxaloacetate; glu, glutamate; αkg, αketoglutarate. (**B**) Plasma enrichment of m+6 glucose during steady-state infusions of [U-13C]glucose. n=3 animals per group. The same color scheme is used throughout this figure. (**C**) Steady-state liver enrichment of m+6 glucose in animals of the indicated genotype. n=3 animals per group. (**D**) Labeled fractions (1-(m+0)) of the indicated metabolites, relative to liver glucose m+6 enrichment. n=3 animals per group. (**E**) Schematic of MRI experiments. Anesthetized animals were challenged to breath room air, followed by 100% O₂, followed by MR imaging to measure perfusion (in room

*Figure 3 continued on next page*

*Figure 3 continued*

air), and blood and tissue oxygenation in each condition. (**F**) Perfusion values (reported as institutional units (I.U.)) based on arterial spin labeling in animals of the indicated genotype. n=3–4 animals per group. (**G**) Relative (%) changes in hepatic T2* following exposure to 100% $O_2$ in animals of the indicated genotype. n=3–4 animals per group. (**H**) Representative heatmaps of T1 apparent values (T1$_{app}$) in livers, overlaid on T1-weighted MR images. (**I**) Representative distribution of T1$_{app}$ values in an animal of each genotype breathing room air, followed by 100% $O_2$. (**J**) Average hepatic T1$_{app}$ values in animals of the indicated genotype, breathing room air followed by 100% $O_2$. n=3–4 animals per group. (**K**) Average regional changes in T1 (ΔT1) in animals of the indicated genotype. n=3–4 animals per group. All data were collected at 4 weeks post AAV-administration. Statistical significance was assessed using two-tailed t-test (**C,D,F,G,K**) or Mann-Whitney (**I**) with adjustments for multiple comparisons. All data represent mean +/-standard deviation from biological replicates. Numerical data for individual panels are provided in ***Supplementary file 1*** and ***Supplementary file 3***.

The online version of this article includes the following figure supplement(s) for figure 3:

**Figure supplement 1.** Metabolic analysis of complex I deficient livers.

assess the contribution of glucose-derived carbons to liver metabolites by measuring the fraction of unlabeled (m+0) metabolite, as well as the fraction of each possible labeled species (m+1, m+2, m+3, …) in which one or more carbons were derived from the labeled glucose molecule. We observed no significant differences in the hepatic enrichment of infused labeled glucose between wild-type and *Ndufa9⁻/⁻* livers (***Figure 3C***). In addition, contribution of glucose-derived carbons to glycolytic and TCA cycle intermediates were unaltered (***Figure 3—figure supplement 1A***), suggesting no substantial changes in glucose handling. As labeling from glucose can be significantly scrambled in vivo and livers exhibit low PDH activity (***Cappel et al., 2019***; ***Merritt et al., 2011***), we also assessed total enrichment (e.g. 1-(m+0)), which revealed no significant differences (***Figure 3D***).

In the conventional view of the mitochondrial ETC, electrons from NADH are transferred via complex I to drive coenzyme Q reduction and $O_2$ consumption via complexes III and IV (***Figure 3A***). In cultured cells, complex I is the major route of electron entry into the ETC, and inhibition of complex I dramatically lowers cellular oxygen consumption (***Figure 1—figure supplement 1A***). Our above results suggest this scenario may not be true during hepatic homeostasis. We therefore made use of magnetic resonance imaging (MRI) to assess blood and tissue oxygenation in vivo (***Figure 3E–K***). Oxygen-sensitive MRI measurements provide a non-invasive method to interrogate changes in oxygen levels in vivo in the absence of a reporter agent. Specifically, the paramagnetic properties of dissolved oxygen within tissue accelerate the decay of longitudinal magnetization of water protons, termed the TOLD (Tissue-Oxygenation-Level-Dependent) effect (***Hallac et al., 2014***). The decay of longitudinal magnetization is characterized by the decay constant (or longitudinal relaxation time) T1. Thus, increased tissue oxygen levels result in decreased T1 values. Similarly, the unpaired electrons of iron in deoxyhemoglobin allow it to act as a strong paramagnetic agent which accelerates the decay of transverse magnetization in the blood (characterized by the transverse relaxation time T2*), termed the BOLD (Blood-Oxygen-Level-Dependent) effect (***Wengler et al., 2019***). In this manner, relative increases in T2* values are indicative of decreased deoxyhemoglobin concentration (i.e., an increased blood oxygenation state). We reasoned that by following hepatic T1 levels in response to increased blood oxygen delivery, we could assess the mitochondria's ability to process this excess oxygen. In these experiments, we controlled for differences in tissue perfusion by measuring tissue perfusion using an arterial spin labeling (ASL) method (***Kim and Tsekos, 1997***). Here, the magnetization of water protons flowing through the slice of interest is labeled by a radio frequency pulse in such a way that they can be distinguished from that of the stationary tissue. The difference in magnetization between the stationary and flowing water protons is related to blood perfusion through a simple two compartment model of tissue, by which the perfusion is quantified (see Materials and methods).

*Ndufa9^f/f* and *Ndufa9⁻/⁻* littermates were challenged with room air, followed by 100% oxygen, and T1 and T2* relaxation parameters were sequentially assessed (***Figure 3E***). Based on arterial spin labeling (***Kim and Tsekos, 1997***), we did not observe changes in liver perfusion between the two genotypes (***Figure 3F***). BOLD (Blood-oxygen-level dependent) measurements (T2*) are dependent on the concentration of deoxyhemoglobin (***Wengler et al., 2019***; ***Barash et al., 2007***), and we observed significant increases in T2* in both *Ndufa9^f/f* and *Ndufa9⁻/⁻* animals, indicating increased blood oxygen in the setting of 100% $O_2$ (***Figure 3G***). TOLD (Tissue-oxygen-level dependent) measurements (based on T1 relaxation) are dependent on the dissolved oxygen concentration; with increased tissue oxygen content, T1 parameters are expected to decrease (***Hallac et al., 2014***). In wild-type livers, we observed no significant decreases in the average hepatic T1 in response to 100% $O_2$ (***Figure 3H, I, J***), consistent

with previous measurements in human livers (*Tadamura et al., 1997*). This suggests that when challenged with increased blood $O_2$ delivery, wild-type mouse liver is able to compensate and maintain steady state $O_2$ levels, potentially through increased tissue oxygen consumption. Similar to *Ndufa9$^{f/f}$* livers, *Ndufa9$^{-/-}$* livers also did not exhibit significant decreases in the average T1 parameter, indicating that tissue oxygen levels are maintained in the absence of complex I function (*Figure 3H–J*). We also assessed T1 changes locally by aligning images (ΔT1); as above, we did not observe significant decreases in the average ΔT1 in either *Ndufa9$^{f/f}$* or *Ndufa9$^{-/-}$* livers (*Figure 3K*). Thus, based on tissue oxygen levels in response to a hyperoxic challenge, *Ndufa9$^{-/-}$* livers do not have a measurable change in oxygen consumption as compared with wild-type livers. Altogether, the above results indicate that complex I is largely dispensable for oxygen status and homeostatic function in the adult mouse liver, without signs of significant metabolic or proteomic compensation.

## Mitochondrial complex IV is required in adult hepatocytes

To further explore a role for the mitochondrial ETC in hepatic function, we examined the requirement for complex IV, making use of a *Cox10$^{flox}$* allele (*Diaz et al., 2005*). *Cox10* is an assembly factor of complex IV, and *Cox10* deletion in skeletal muscle is associated with diminished cIV activity accompanied by myopathy (*Diaz et al., 2005*). A previous report utilizing this allele in combination with the liver-specific Albumin-Cre transgenic reporter revealed partially penetrant phenotypes dependent on the transgene dosage (*Diaz et al., 2008*), suggesting that the mitochondrial ETC is required for liver development. Thus, we made use of AAV-delivery of Cre recombinase to *Cox10$^{f/f}$* animals to investigate its role in the adult liver. Similar to above, we injected *Cox10$^{f/f}$* animals with AAV-GFP or AAV-Cre and characterized liver physiology at 4 and 8 weeks post-injection. At 4 weeks post-injection, AAV-Cre injected animals (hereafter, '*Cox10$^{-/-}$*') demonstrated significant reductions in *Cox10* mRNA and protein (*Figure 4A and B*) and complex IV activity (*Figure 4C*). By 8 weeks post-AAV administration, *Cox10$^{-/-}$* livers were grossly enlarged and pale in color, as compared with wild-type livers (*Figure 4D and F*). Histological analysis of liver sections revealed signs of lipid accumulation indicating hepatic steatosis (Oil Red O staining), as well as decreased glycogen accumulation (PAS staining) (*Figure 4E*). Fasting glucose levels were decreased in animals with *Cox10$^{-/-}$* livers (*Figure 4G*). To directly probe gluconeogenesis, fasted mice were challenged with a [U-$^{13}$C]-lactate/pyruvate tolerance test; animals with *Cox10$^{-/-}$* livers were unresponsive as compared with wild-type animals (*Figure 4H, I*, *Figure 4— figure supplement 1A*). Additional plasma indicators of liver function were diminished in *Cox10$^{-/-}$* animals, including decreased plasma total protein and increased plasma bilirubin (*Figure 4J*). We also observed significant elevations in ALT and AST, indicators of liver damage (*Figure 4K*). Thus, in contrast to loss of complex I, the loss of complex IV in adult hepatocytes is associated with hepatic steatosis, diminished liver function and evidence of liver damage.

Despite the fatty livers and diminished function, *Cox10$^{-/-}$* animals exhibited no survival deficits even 8 weeks after AAV administration, and we did not detect changes in steady state ATP levels in liver tissue (*Figure 5A*). *Cox10$^{-/-}$* livers exhibited a small, but not statistically significant, trend towards increased oxidative stress (lowering of the GSH / GSSG ratio) (*Figure 5B*). In contrast to cI-deficient livers, cIV deficiency significantly impacted the redox status of the cell, as measured by the NAD$^+$ / NADH ratio (*Figure 5C*). We observed altered ratios of both pyruvate / lactate and acetoacetate / β-hydroxybutyrate, indicating impaired redox status in both the cytosol and mitochondrial compartments (*Figure 5D*). Interestingly, tissue α-hydroxybutyrate levels were decreased in *Cox10$^{-/-}$* livers, and plasma α-hydroxybutyrate levels were not significantly affected (*Figure 5E*). Thus, impairment of mitochondrial complex IV has both biochemical and functional consequences in the adult liver and is required for maintenance of cellular redox status.

To probe the effects of complex IV loss in mammalian liver, we assessed proteome changes by label-free proteomics (*Figure 5F*, *Supplementary file 2*). Unsupervised hierarchical clustering was able to separate proteomes between *Cox10$^{f/f}$* and *Cox10$^{-/-}$* livers (*Figure 5F*). A large number of proteins were significantly up-regulated ($\log_2$(Fold Change)>1; p-value <0.05), and this group was enriched (209 out of 319) for proteins that localize to mitochondria (*Figure 5G* (red dots), *Supplementary file 2*). Gene ontology analysis of all upregulated proteins revealed enrichment of a large number of mitochondrial pathways in all three categories (Biological Processes (BP), Cellular Components (CC), and Molecular Functions (MF)) (*Supplementary file 2*, *Figure 5—figure supplement 1A* (red arrows)). Consistent with this, there were significant increases in total mitochondrial content in *Cox10$^{-/-}$* livers,

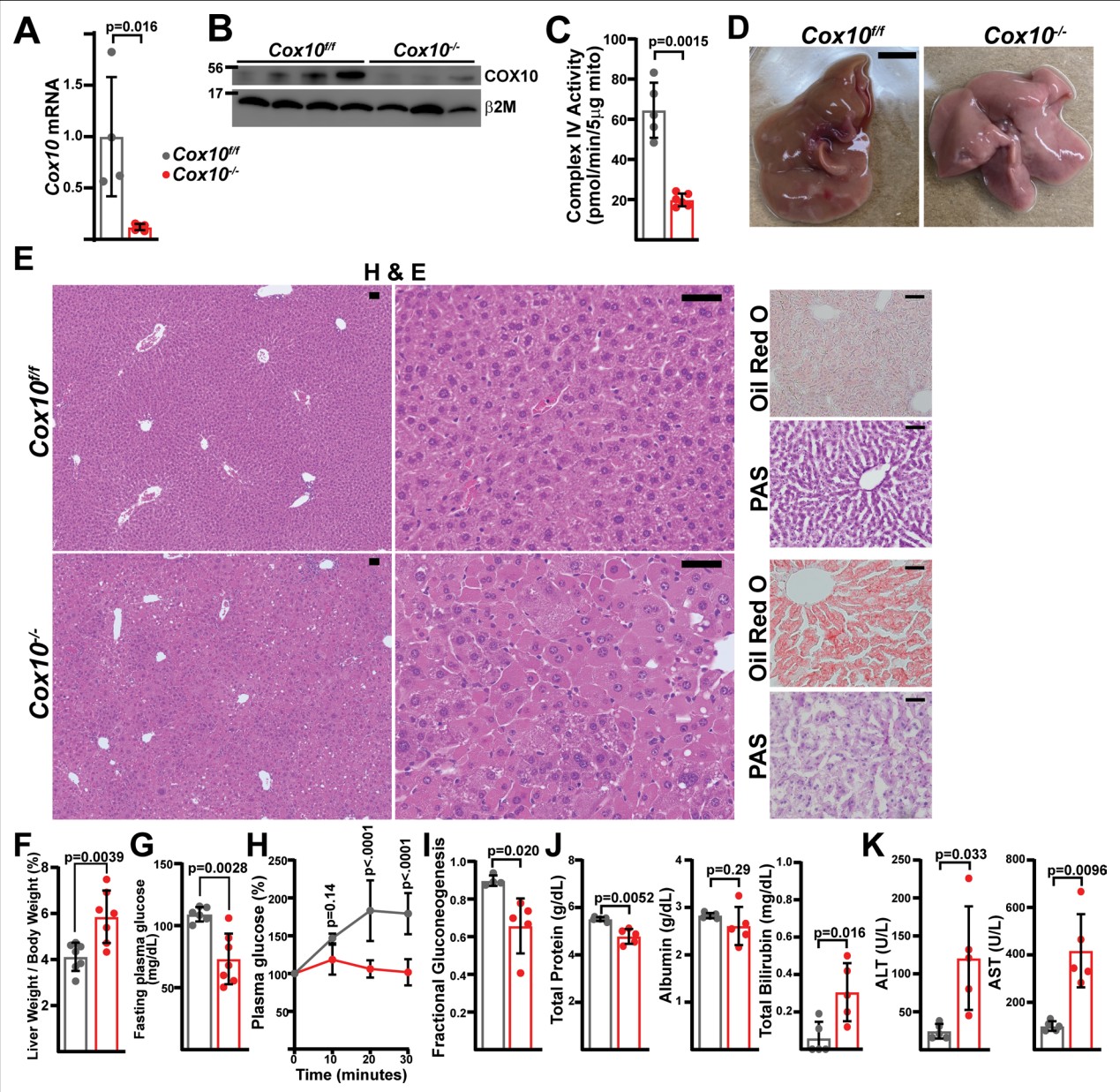

**Figure 4.** Complex IV deficiency in adult hepatocytes induces liver dysfunction. (**A**) *Cox10* mRNA levels (relative to *β2-microglobulin*; normalized) in livers of the indicated genotype, assessed by qRT-PCR at 4 weeks post-AAV administration. n=4–5 animals per group. The same color scheme is used throughout this figure. (**B**) Cox10 protein levels in livers of the indicated genotype, assessed by Western blot at 4 weeks post AAV-administration. β2-microglobulin levels are shown as a loading control. MW markers are indicated in kDa. (**C**) Complex IV activity (pmol O₂ consumed / min) in isolated mitochondria supplemented with ascorbate/TMPD substrates from livers of the indicated genotype, assessed at 4 weeks post AAV-administration. n=5–6 animals per group. (**D**) Representative images of gross liver anatomy in animals of the indicated genotype at 8 weeks post AAV-administration. Scale bar: 0.5 cm. (**E**) Representative histology (H&E, Oil Red O, PAS staining) images of liver cross sections from animals of the indicated genotype at 8 weeks post-AAV administration. Scale bar: 50 μm. (**F**) Liver weight (normalized to body weight) from mice of the indicated genotype at 8 weeks post-AAV administration. n=7 animals per group. (**G**) Fasting plasma glucose levels in mice of the indicated genotype at 8 weeks post-AAV administration. n=6–7 animals per group. (**H**) Relative changes in plasma glucose following a lactate/pyruvate tolerance test in mice of the indicated genotype at 8 weeks post-AAV administration. n=4–5 mice per group. (**I**) Fractional gluconeogenesis (plasma) in animals of the indicated genotype, measured via pyruvate/lactate tolerance test at 8 weeks post-AAV administration. n=4–5 animals per group. (**J**) Circulating plasma markers (total protein, albumin, and total bilirubin) of liver function in mice of the indicated genotype at 8 weeks post-AAV administration. n=5 animals per group. (**K**) Plasma markers (ALT and AST) of liver damage in mice of the indicated genotype at 8 weeks post-AAV administration. n=5 mice per group. Statistical significance was assessed using two-way ANOVA (**H**), t-test (**C,F,G,I,J,K**) or Mann-Whitney (**A,K**) tests with adjustments for multiple comparisons. All data represent mean

*Figure 4 continued on next page*

*Figure 4 continued*

+/-standard deviation from biological replicates. Full gel images are provided in *Figure 4—source data 1*. Numerical data for individual panels are provided in *Supplementary file 3*.

The online version of this article includes the following source data and figure supplement(s) for figure 4:

**Source data 1.** Full gel images for *Figure 4*.

**Figure supplement 1.** Lactate/pyruvate tolerance test in complex IV deficient livers.

as measured by both mitochondrial genome and protein content (*Figure 5I–K*; *Figure 5—figure supplement 8*). Gene ontology analysis of mitochondrial proteins indicated enrichment of several gene groups, including electron transport, TCA cycle, translation, and ATP synthesis (*Supplementary file 2*, *Figure 5—figure supplement 1B*). A separate analysis focused on non-mitochondrial proteins that were upregulated did not reveal many pathways that were significantly enriched (FDR <0.05), with the single exception of the cornified envelope cellular component pathway, which included various genes implicated in cell damage and death processes [e.g. S100A9 (*Ghavami et al., 2010*; *Ghavami et al., 2008*), LGALS3 (*Mazurek et al., 2012*), NCL (*Kobayashi et al., 2012*; *Goldstein et al., 2013*; *Yang et al., 2002*), HSPB1 (*Singh et al., 2017*)] (*Supplementary file 2*, *Figure 5—figure supplement 1C*), and may potentially be related to the hepatic damage observed in this animal model (*Figure 4K*).

Only a small number of mitochondrial proteins were present among the downregulated proteins (16 of 214), among which we identified a number of complex IV components (*Figure 5G and H*), consistent with destabilization of intact complex IV due to loss of an assembly subunit. We did not observe significant decreases in components from other ETC complexes, suggesting an isolated effect of *Cox10* loss on complex IV (*Figure 5H*). Consistent with this, gene ontology analysis of all downregulated proteins identified complex IV as the only Cellular Components pathway enriched among the downregulated genes (*Supplementary file 2*, *Figure 5—figure supplement 2A*). Several biological processes related to metabolism were enriched among the downregulated genes which included lipid metabolic and biosynthetic pathways (*Supplementary file 2*, *Figure 5—figure supplement 2A* (blue arrows)). Analyses focusing specifically on mitochondrial and non-mitochondrial downregulated genes revealed similar findings (i.e. downregulation of mitochondrial complex IV components, as well as metabolic processes) (*Supplementary file 2*, *Figure 5—figure supplement 2B*,C). Thus, proteomic changes in response to hepatic complex IV deficiency largely center around increased mitochondrial constituents, downregulation of complex IV, as well as decreased enzyme abundance for a number of metabolic pathways.

To further characterize the compensatory response in the absence of complex IV, we performed label-free proteomics on enriched mitochondrial fractions from $Cox10^{f/f}$ and $Cox10^{-/-}$ livers, which was able to identify 721 mitochondrial proteins (*Supplementary file 2*). We observed significant changes ($|\log_2(\text{fold change})|>1$, Adjusted p-val <0.05) in 99 proteins (~14% of detected mitochondrial proteins) (*Figure 5—figure supplement 3A*, *Supplementary file 2*). Complex IV subunits tended to be decreased in abundance, and we did not observe significant changes in most subunits of the other complexes although they tended to be upregulated (*Figure 5—figure supplement 3B*). We observed statistically significant downregulation of some select components of fatty acid metabolism (FABP1, ACAD11) as well as upregulation of some components regulating ubiquinol homeostasis (DHODH, UQCC1, UQCC2) (*Figure 5—figure supplement 3A*).

Steady-state metabolomic analysis of hepatic tissue revealed a number of metabolites altered in response to complex IV loss, and unsupervised hierarchical clustering was sufficient to separate $Cox10^{f/f}$ and $Cox10^{-/-}$ livers (*Figure 5L and M*). Mitochondrial TCA cycle metabolites were largely maintained, although we observed a significant increase in 2-hydroxyglutarate levels, which has been previously observed in the setting of mitochondrial dysfunction (*Figure 5N*; *Hunt et al., 2019*). We observed significant upregulation in a number of cholate species, consistent with cholestasis and hepatomegaly in these animals (*Figure 5M*, *Figure 5—figure supplement 4A*). Upregulated metabolites were also highly enriched for long-chain acylcarnitine fatty acid species (*Figure 5M*). We hypothesize that the accumulation of acylcarnitine species is related to a fatty oxidation defect secondary to loss of mitochondrial complex IV; however, this finding may also be related to the proteomic changes in $Cox10^{-/-}$ livers discussed above. We therefore assessed proteomic changes in annotated GO pathways related to fatty acid metabolism (*Figure 5—figure supplement 5*). A number of enzymes related to fatty

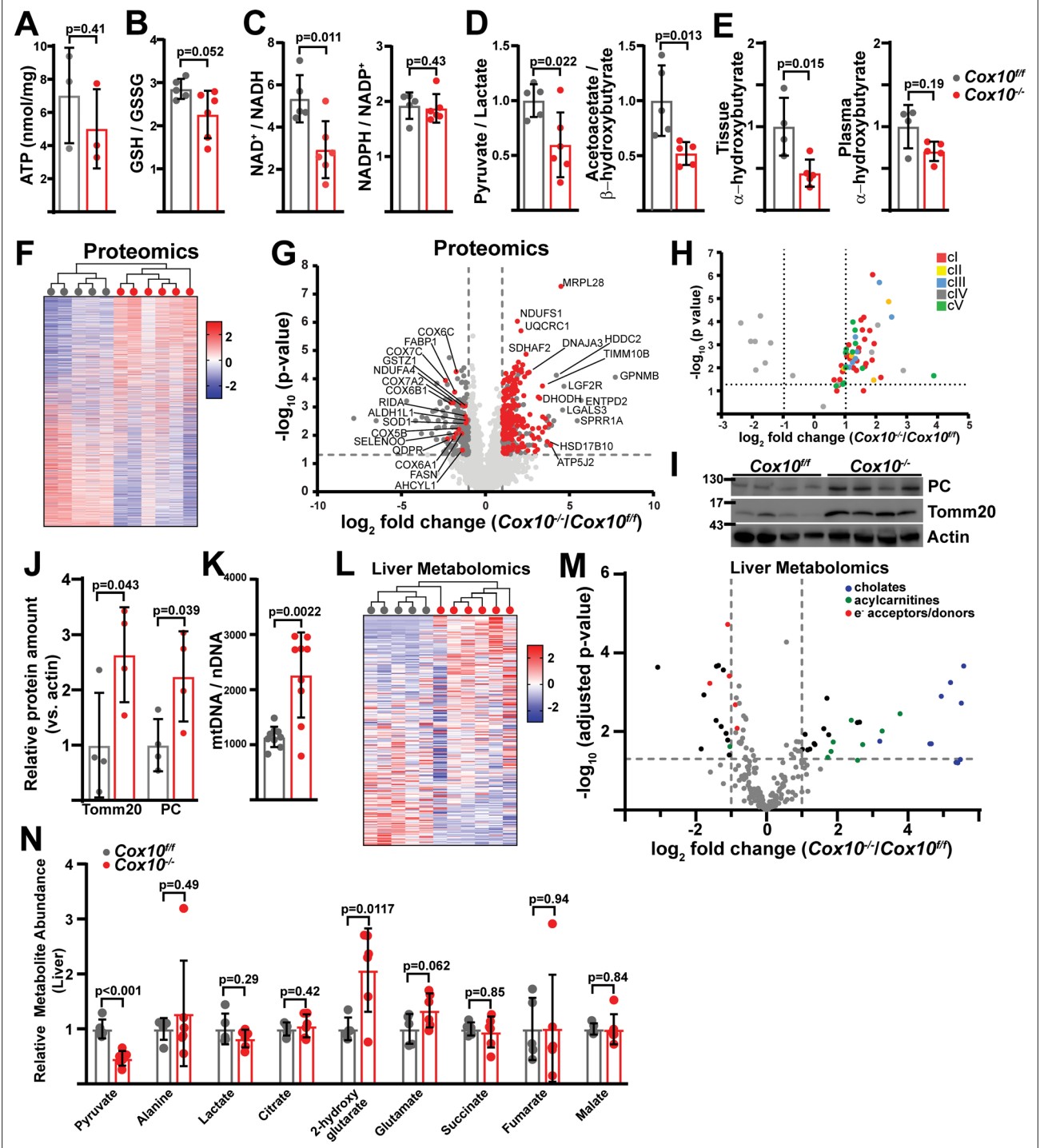

**Figure 5.** Complex IV deficiency alters metabolism and cellular redox status in adult liver. (**A**) ATP levels in livers of the indicated genotype. n=3 animals per group. The same color scheme is used throughout this figure. (**B**) Reduced glutathione to oxidized glutathione ratio (GSH:GSSG) in livers of the indicated genotype, as a proxy of oxidative stress. n=5–6 mice per group. (**C**) NAD⁺/NADH and NADPH/NADP⁺ ratios in livers of the indicated genotype. n=5–6 mice per group. (**D**) Normalized cytosolic (pyruvate / lactate) and mitochondrial (acetoacetate / β–hydroxybutyrate) redox ratios in livers of the indicated genotype. n=5–6 mice per group. (**E**) Normalized tissue and plasma α–hydroxybutyrate levels in animals of the indicated genotype. n=4–5 mice per group. (**F**) Unsupervised hierarchical clustering of protein abundances in *Cox10^{f/f}* and *Cox10^{-/-}* livers, depicted as a heatmap of z-scores. (**G**) Volcano plot of protein abundances changes in *Cox10^{-/-}* vs. *Cox10^{f/f}* livers, based on proteomics analysis. Log₂(Fold change) is plotted against the –log₁₀(adjusted p-value) for each protein. Significantly changing proteins (log₂(Fold change)>1 or < –1; Adjusted p-val <0.05) are colored dark gray; of these, mitochondria-localized proteins are colored red. n=5–6 mice per group. (**H**) Volcano plot of abundances for mitochondrial ETC

*Figure 5 continued on next page*

*Figure 5 continued*

proteins in *Cox10⁻/⁻* vs. *Cox10^f/f* livers, based on proteomics analysis. (**I**) Levels of mitochondrial proteins (PC, pyruvate carboxylase, and Tomm20) in livers of the indicated genotype, as assessed by western blot. Actin levels are shown as a loading control. MW markers are indicated in kDa. (**J**) Relative amounts of Tomm20 and PC in livers of the indicated genotype, relative to actin levels. n=4 animals per group. (**K**) Mitochondrial genome (mtDNA) to nuclear genome (nDNA) ratios in livers of the indicated genotype. n=9 mice per group. (**L**) Unsupervised hierarchical clustering of metabolite abundances in *Cox10^f/f* and *Cox10⁻/⁻* livers, depicted as a heatmap of z-scores. (**M**) Volcano plot of metabolite abundance changes in *Cox10⁻/⁻* vs. *Cox10^f/f* livers, based on metabolomics analysis. $\log_2$(Fold change) is potted against the $-\log_{10}$(adjusted p-value) for each metabolite. Significantly changing metabolites ($\log_2$(Fold change)>1 or < −1; Adjusted p-val <0.05) are colored black. n=5–6 mice per group. (**N**) Relative abundances of mitochondrial TCA cycle and related metabolites from livers of the indicated genotype. n=5–6 mice per group. All data in this figure were collected at 8 weeks post AAV-administration. Statistical significance was assessed using t-test (**A,B,C,D,E,J,K,N**), or Mann-Whitney (**C,E**) tests with adjustments for multiple comparisons. All data represent mean +/-standard deviation from biological replicates. Full gel images are provided in *Figure 5—source data 1*. Numerical data for individual panels are provided in *Supplementary file 2* and *Supplementary file 3*.

The online version of this article includes the following source data and figure supplement(s) for figure 5:

**Source data 1.** Full gel images for *Figure 5*.

**Figure supplement 1.** Functional annotation (DAVID) analysis of upregulated proteins in complex IV deficient livers.

**Figure supplement 2.** Functional annotation (DAVID) analysis of downregulated proteins in complex IV deficient livers.

**Figure supplement 3.** Analysis of mitochondrial proteomes in complex-IV-deficient livers.

**Figure supplement 4.** Metabolic analysis of complex-IV-deficient livers.

**Figure supplement 5.** Metabolic / proteomic analysis of fatty-acid-related pathways in complex-IV-deficient livers.

**Figure supplement 6.** Metabolic / proteomic analysis of bile-acid-related pathways in complex-IV-deficient livers.

**Figure supplement 7.** Metabolic / proteomic analysis of oxidoreductase-related pathways in complex-IV-deficient livers.

**Figure supplement 8.** Immunofluorescence of hepatic sections.

---

acid synthesis exhibited statistically significant down regulation in *Cox10⁻/⁻* livers, including FASN, ACLY, and ACSL1, key fatty acid synthesis enzymes (*Figure 5—figure supplement 5A*,C). However, we did not observe overall enrichment or depletion of the fatty acid synthesis pathway by gene set enrichment analysis (GSEA) (*Figure 5—figure supplement 5A*), and we did not detect significant changes in ACACA and ACACB, which catalyze rate-limiting steps for fatty acid synthesis. Conversely, genes involved in fatty acid oxidation tended to be upregulated (*Figure 5—figure supplement 5B*,C), although the overall enrichment of this pathway did not reach statistical significance, and we did not observe significant changes in the rate-limiting enzyme CPT1A. As a whole, the trends towards down-regulation of fatty acid synthesis enzymes and upregulation of fatty acid oxidation enzymes are not consistent with being causal for the large accumulation of acyl-carnitine species, and instead suggest that the proteome responds in a compensatory manner to limit the accumulation of acyl-carnitine species.

We similarly examined proteomic changes in genes involved in bile acid metabolism (*Figure 5—figure supplement 6*), and their relation to the large increases in levels of cholate species (*Figure 5M*, *Figure 5—figure supplement 4A*). Abundance of these family members tended to be decreased in *Cox10⁻/⁻* livers, although the overall pathway depletion did not reach statistical significance (*Figure 5—figure supplement 6A*). Unfortunately, our proteomic analysis did not detect most bile acid metabolic enzymes, including the rate-limiting enzyme CYP7A1 (*Figure 5—figure supplement 6B*).

Downregulated metabolites were enriched for several electron carriers, including pyruvate, ascorbate and tetrahydrobiopterin (*Figure 5M*, *Figure 5—figure supplement 4B*). Pyruvate can exchange electrons with NAD(H), while ascorbate and THBP are potent antioxidants which can exchange electrons with metal ions and hydroxyradical species (*Traber and Stevens, 2011*; *Kojima et al., 1995*). Thus, inhibition of the mitochondrial ETC via *Cox10* removal may influence redox balance in multiple species. To assess related proteomic changes, we investigated alterations in genes of the oxidoreductase family, a large family related to transfer of electrons between many potential substrates (GO: 0016491; 787 members). Overall, oxidoreductase enzymes trended to be upregulated in *Cox10⁻/⁻* livers, largely driven by mitochondrial-localized proteins, and we did identify a number of enzymes related to pyruvate, ascorbate and THBP metabolism as significantly altered in response to hepatic complex IV loss (*Figure 5—figure supplement 7A*). The overall pathway enrichment did not reach statistical significance. Focusing on each electron carrier in isolation did reveal some interesting trends. Pyruvate has a number of potential metabolic fates, and we did observe upregulation of

components of the PDH complex (PDHA1, PDHB, PDHX, DLAT, DLD) which could potentially deplete pyruvate via conversion into acetyl-CoA (*Figure 5—figure supplement 7B*). Similarly, we observed significant upregulation of BCKDHA and BCKDHB, which potentially serve to deplete the electron carrier α-hydroxybutyrate; in addition, the rate-limiting enzyme of this transsulfuration pathway (CBS) was significantly downregulated (*Figure 5—figure supplement 7C*). The ketone metabolites aceto-acetate and β-hydroxybutyrate were significantly depleted in *Cox10*$^{-/-}$ livers; however, we did not observe significant changes in the ketone body-related enzymes detected in our dataset, including the rate-limiting synthesis enzymes HMGCS1 and HMGCS2 (*Figure 5—figure supplement 7D*). With respect to ascorbate metabolism, the majority of related enzymes were not detected in our experiment, including the rate-limiting synthesis enzyme GULO (*Figure 5—figure supplement 7E*). Lastly, we detected a significant decrease in the BH4 (tetrahydrobiopterin) recycling enzyme QDPR, consistent with the downregulation of BH4 levels. The rate-limiting synthesis enzyme, GCH1, was not detected. To summarize, the precise reasons for changes in the levels of these electron carriers is not clear at this stage; however, it is possible that proteome changes in response to complex IV loss may be contributing to these alterations.

## Complex IV inhibition blocks electron entry from the electron transfer flavoprotein (ETF) complex

In cultured cells, Complex IV is the major source of oxygen consumption. To examine its role in vivo, we performed magnetic resonance imaging in *Cox10*$^{f/f}$ and *Cox10*$^{-/-}$ animals as described above (*Figure 3E*). Based on arterial spin labeling measurements, we observed no alterations in perfusion of *Cox10*$^{-/-}$ livers (*Figure 6A*). In addition, BOLD (T2*) measurements indicated significant increases in blood oxygen content in both genotypes in response to 100% $O_2$, based on increases in the T2* relaxation time (*Figure 6B*). Similar to above results, *Cox10*$^{f/f}$ (wild-type) livers did not display increases in tissue oxygen levels, based on TOLD imaging and average T1 parameters (*Figure 6C–E*). In contrast, *Cox10*$^{-/-}$ livers displayed significant decreases in average T1 relaxation times in response to a 100% $O_2$ challenge, indicating increased dissolved oxygen levels in affected liver tissue (*Figure 6C–E*). A regional assessment of changes in T1 (ΔT1) also indicated decreased T1 parameters in *Cox10*$^{-/-}$, but not *Cox10*$^{f/f}$ livers (*Figure 6F*). Thus, *Cox10*$^{-/-}$ livers are deficient in maintaining oxygen levels in response to increased blood $O_2$ delivery, consistent with decreased oxygen consumption in the setting of cIV deficiency.

The contrast in the tissue oxygen status between *Ndufa9*$^{-/-}$ and *Cox10*$^{-/-}$ livers suggests that complex I is not a significant electron donor to the hepatic ETC, at least under homeostatic conditions. If complex I was a major electron source, we would expect that depletion of complex IV might result in altered glucose utilization in the mitochondria. We therefore performed steady-state euglycemic infusions of [U-$^{13}$C]glucose in *Cox10*$^{f/f}$ and *Cox10*$^{-/-}$ animals (*Figure 6G*). Similar to *Ndufa9*$^{-/-}$ animals, we did not observe altered glucose handling in *Cox10*$^{-/-}$ animals (*Figure 6H, I*; *Figure 6—figure supplement 1A*), suggesting that complex I activity is low and glucose utilization in the mitochondria is not sensitive to ETC function.

In principle, there are several paths of electron entry (*Figure 6J*) independent of complex I, including complex II (which receives electron from succinate), DHODH (which receives electrons from dihydroorotate), GPD2 (which receives electrons from glycerol 3-phosphate), and the electron transfer flavoprotein (ETF) complex (which receives electrons from FADH$_2$). These alternate routes bypass complex I by donating electrons directly to ubiquinone. Complex IV dysfunction is expected to impact all of these routes; however, we would only expect to see significant metabolic alterations in pathways with high flux. In culture, ETC inhibition is commonly associated with elevated succinate levels; however, we did not observe this result in *Cox10*$^{-/-}$ livers (*Figure 6K*). Similarly, we did not observed elevations in glycerol 3-phosphate or dihydroorotate (*Figure 6K*), suggesting that GPD2 and DHODH activities are not a major electron source in murine livers.

Lastly, we examined whether the ETF complex constitutes a significant electron source to the hepatic ETC. The ETF complex is made up of three proteins (ETFA, ETFB and ETFDH), which transfers electrons from FADH$_2$ to ubiquinone. Significant sources of FADH$_2$ are thought to include fatty acids (via beta-oxidation), as well as amino acid metabolism (*Figure 6J*). In *Cox10*$^{-/-}$ livers, we observed significant elevations in medium/long chain acyl-carnitines (*Figure 6L*), as well as a decrease in β-hydroxybutyrate levels (*Figure 6M*), consistent with a fatty acid oxidation defect. In addition, we

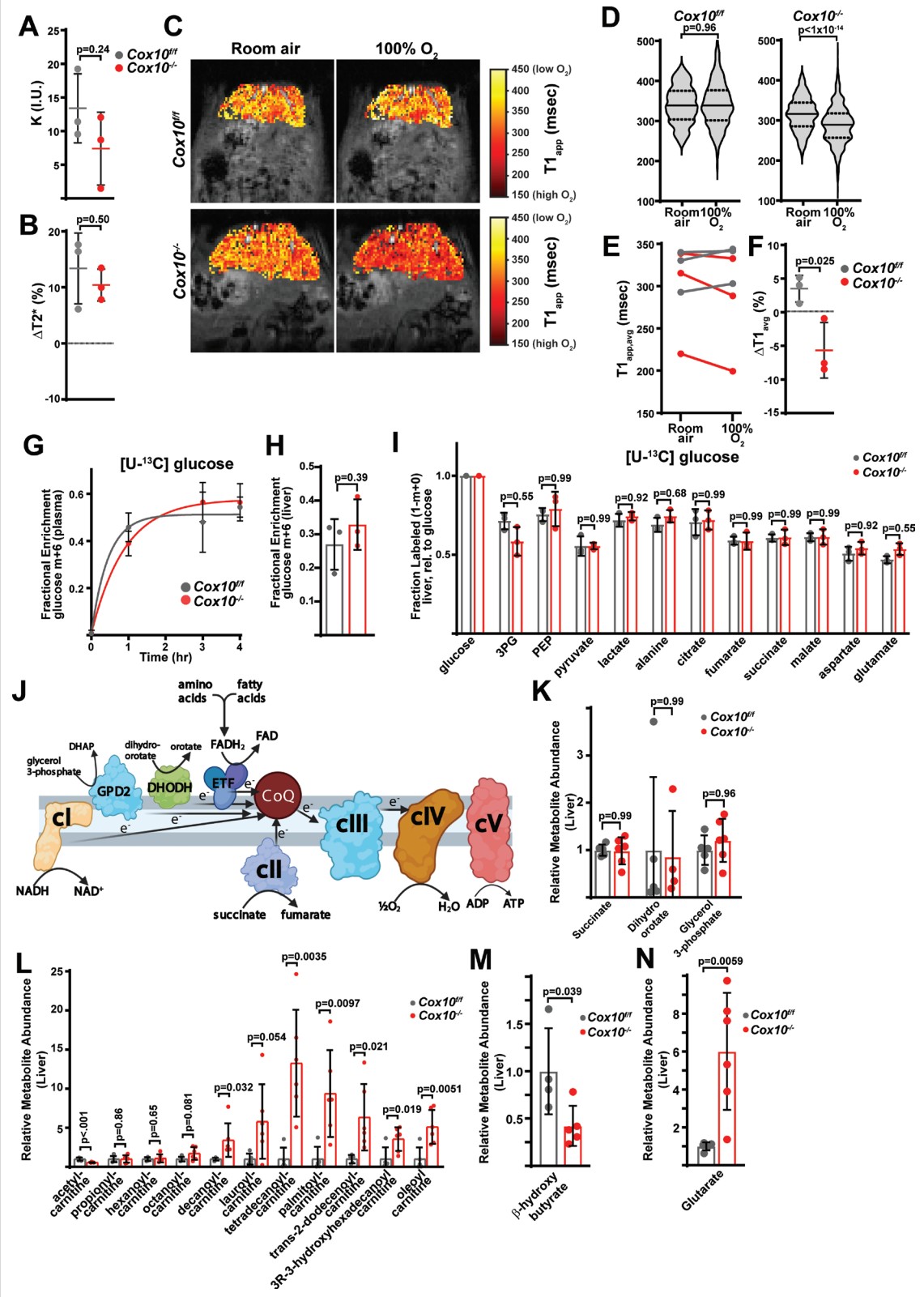

**Figure 6.** Complex IV deficiency impairs electron transport chain function and oxygen status in the adult liver. (**A**) Perfusion values (reported as institutional units (I.U.)) based on arterial spin labeling in animals of the indicated genotype at 4 weeks post-AAV administration. n=3 animals per group. The same color scheme is used throughout this figure. (**B**) Relative (%) changes in hepatic T2* following exposure to 100% $O_2$ in animals of the indicated genotype at 4 weeks post-AAV administration. n=3 animals per group. (**C**) Representative heatmaps of T1 apparent values (T1$_{app}$) in livers,

*Figure 6 continued on next page*

*Figure 6 continued*

overlaid on T1-weighted MR images. (**D**) Representative distribution of $T1_{app}$ values in an animal of each genotype breathing room air, followed by 100% $O_2$. (**E**) Average hepatic $T1_{app}$ values in animals of the indicated genotype, breathing room air followed by 100% $O_2$. n=3 animals per group; 4 weeks post-AAV administration. (**F**) Average regional changes in T1 (ΔT1) in animals of the indicated genotype. n=3 animals per group; 4 weeks post-AAV administration. (**G**) Plasma enrichment of m+6 glucose during steady-state infusions of [U-$^{13}$C]glucose. n=3 animals per group; 4 weeks post-AAV administration. (**H**) Steady-state liver enrichment of m+6 glucose in animals of the indicated genotype. n=3 animals per group; 4 weeks post-AAV administration. (**I**) Labeled fractions (1-(m+0)) of the indicated metabolites, relative to liver glucose m+6 enrichment. n=3 animals per group; 4 weeks post-AAV administration. (**J**) Schematic of potential electron sources for mitochondrial ETC. CoQ, ubiquinone; GPD2, mitochondrial glycerol-3-phosphate dehydrogenase; DHODH, dihydroorotate dehydrogenase. ETF, electron transfer flavoprotein complex. DHAP, dihydroxyacetone phosphate. (**K**) Relative abundance of electron donors in livers of the indicated genotype at 8 weeks post-AAV administration. n=4–6 mice per group. (**L**) Relative abundance of acylcarnitine metabolites in livers of the indicated genotype. n=5–6 mice per group at 8 weeks post-AAV administration. (**M**) Relative β-hydroxybutyrate levels in livers of the indicated genotype at 8 weeks post-AAV administration. n=4–5 mice per group. (**N**) Relative glutarate levels in livers of the indicated genotype at 8 weeks post-AAV administration. n=5–6 mice per group. Statistical significance was assessed using t-test (**A,B,E,F,H,I,K,L,M,N**) or Mann-Whitney (**D**) tests with adjustments for multiple comparisons. All data represent mean +/-standard deviation from biological replicates. Numerical data for individual panels are provided in *Supplementary file 2* and *Supplementary file 3*.

The online version of this article includes the following figure supplement(s) for figure 6:

**Figure supplement 1.** Isotope tracing in complex-IV-deficient livers.

observed a striking increase in glutarate levels (*Figure 6N*), consistent with a selective defect in amino acid metabolism linked to $FADH_2$. Thus, the metabolic abnormalities observed in *Cox10*$^{-/-}$ livers are most consistent with a buildup of ETFDH substrates.

## Discussion

The dispensability of complex I for homeostatic function in the adult murine liver is not consistent with a key role for cI in electron entry to the ETC in this tissue; in particular, we observed no alterations in mitochondrial or cytosolic redox status in cI-deficient liver, or evidence of significant metabolic, histologic or functional alterations. It is possible that cI-related defects are minor and not detectable by our assays. The mouse liver is known to have relatively low flux through PDH, the major entry point of carbons to the TCA cycle which generates NADH for cI function (*Cappel et al., 2019*; *Merritt et al., 2011*); thus, our data is consistent with a model in which the hepatic TCA cycle does not play a prominent role in nutrient oxidation via the generation of mitochondrial NADH under homeostatic conditions. The impact of cIV deficiency on cellular redox status indicates that the mitochondrial ETC is critical for removing electrons via transfer to oxygen, and thus, alternative electron donors are likely to fuel the ETC. However, the lack of a survival deficit in *Cox10*$^{-/-}$ animals in this study calls into question the absolute requirement for an intact mitochondrial ETC in mouse liver.

A precise measurement of the source of electrons for the ETC is challenging due to an inability to directly track electrons after incorporation into the redox-sensitive cofactors used by the mitochondrial ETC complexes. Our results from complex IV dysfunctional livers reveal a buildup of substrates that normally feed into the ETF complex, including elevations in multiple acyl-carnitine species, as well as glutarate, a byproduct of lysine and tryptophan metabolism. In principle, the buildup of acyl-carnitines could be due to increased rates of fatty acid synthesis (FAS); however, we observed that a number of FAS enzymes were downregulated. In addition, the reduction in β-hydroxybutyrate (a terminal product of fatty acid oxidation) levels combined with the large increases in fatty acid species is consistent with a defect in fatty acid oxidation (FAO), although these steady state measurements do not directly inform on flux through FAO pathways, and experiments to directly trace flux through the ETF complex will be necessary.

In humans, mutations in proteins of the ETF complex are causative for glutaric acidemia type II disease (also known as multiply acyl-CoA dehydrogenase deficiency), which has similar biochemical abnormalities as our *Cox10*$^{-/-}$ animal model (*Frerman and Goodman, 1985*; *Grünert, 2014*). The clinical progression of these patients can be variable; however, patients with hepatic involvement exhibit hepatic steatosis, elevated ALT/AST levels and periodic hypoglycemia (*Siano et al., 2021*). In addition, mice carrying a knock-in ETFDH mutation displayed diet-dependent phenotypes, including hepatic

steatosis and elevated levels of acyl-carnitine species (*Xu et al., 2018*). Lastly, patients with SCO1 mutations (impacting complex IV assembly) display hepatic steatosis and elevated acyl-carnitines (*Valnot et al., 2000*; *Stiburek et al., 2009*), suggesting a clinical overlap between mitochondrial ETC inhibition and ETF complex deficiency.

Our results therefore indicate the complex I is dispensable in the adult murine liver. A limitation of our study is that we have focused on steady state analysis of homeostatic functions in the mouse liver of the C57BL/6 J background, and it is possible that complex I does have a required role in alternative genetic backgrounds, or in the setting of hepatic stressors or alternative diets. Indeed, it has been reported that C57BL/6 J animals contain a mutation which prevents the incorporation of multimeric complex IV into supercomplexes, and so the major contributing sources of electrons for the ETC may be different in this background (*Lapuente-Brun et al., 2013*; *Jha et al., 2016*). Thus, it is possible that cI is required in other genetic backgrounds and it will be important to assess these scenarios in future work. Indeed, the relevant and high flux modes of electron entry into the ETC may be different among distinct tissues and backgrounds, and this provides a plausible hypothesis for the non-overlapping syndromes present in mitochondrial ETC diseases affecting different components. As an upstream component of the ETC, cI mutations from the mitochondrial or nuclear genome may manifest in a narrow set of tissues systems which rely on mitochondrial NADH to fuel respiration. In contrast, complex IV, the terminal component of the ETC which feeds electrons to oxygen as a final electron acceptor, might be predicted to have effects in a wider variety of tissues, particularly those which do not have alternative pathways to remove excess electrons. Understanding the key ETC fuel sources in both normal and stressed tissue systems will therefore provide insight into the pathophysiological processes which occur downstream of specific mitochondrial mutations.

# Materials and methods

## Key resources table

| Reagent type (species) or resource | Designation | Source or reference | Identifiers | Additional information |
|---|---|---|---|---|
| Strain, strain background (*Mus musculus*, male and female) | *Cox10^f/f* | The Jackson Laboratory | 024697 | |
| Strain, strain background (*Mus musculus*, male and female) | *Ndufa9^f/f* | | | |
| Other | pAAV.TBG.PI.eGFP. WPRE.bGH (AAV-GFP) | Addgene | 105535 | Adeno-associated virus |
| Other | pAAV.TBG.PI.Cre. rBG (AAV-Cre) | Addgene | 107787 | Adeno-associated virus |
| Antibody | Anti-Tom20 (rabbit polyclonal) | Proteintech | 11802 | WB, 1:2000 |
| Antibody | Anti-PC (rabbit polyclonal) | Proteintech | 16588 | WB, 1:1000 |
| Antibody | Anti-βActin (mouse monoclonal) | Proteintech | 66009 | WB, 1:5000 |
| Antibody | Anti-Ndufa9 (mouse monoclonal) | ThermoScientific | 459100 | WB, 1:2000 |
| Antibody | Anti-β2microglobulin (rabbit monoclonal) | ThermoScientific | 701250 | WB,:5000 |
| Antibody | Anti-Cox10 (rabbit polyclonal) | Abcam | ab84053 | WB,:1000 |
| Commercial assay or kit | Complex I Enzyme Activity Microplate Assay Kit (Colorimetric) | Abcam | ab109721 | |
| Commercial assay or kit | ATP Colorimetric/ Fluorometric Assay Kit | Biovision | K354 | |

*Continued on next page*

*Continued*

| Reagent type (species) or resource | Designation | Source or reference | Identifiers | Additional information |
|---|---|---|---|---|
| Commercial assay or kit | Deproteinizing Sample Preparation Kit | Biovision | K808 | |
| Commercial assay or kit | Luna Universal One-Step RT-qPCR kit | New England Biolabs | E3005S | |
| Commercial assay or kit | DC Protein Assay | Biorad | 5000112 | |
| Chemical compound, drug | [U-$^{13}$C]glucose | Cambridge Isotopes Laboratories | CLM-1396 | |
| Chemical compound, drug | [U-$^{13}$C]lactate | Cambridge Isotopes Laboratories | CLM-1579 | |
| Chemical compound, drug | [U-$^{13}$C]pyruvate | Cambridge Isotopes Laboratories | CLM-2440 | |

## Reagents

Most chemicals and reagents including antimycin A (A8674), carbonyl cyanide 3-chlorophenylhydrazone [CCCP (C2759)], EGTA (E3889), glutamate (49621), glycerol (G5516), hematoxylin (GHS132), HEPES (H4034), magnesium chloride (208337), malate (240176), D-mannitol (M4125), NAD$^+$ (N8285), NADP$^+$ (N8160), NADH (N6785), NADPH (N9960), $^{15}$N$_5$-AMP (900382), Orotic acid (O2750), L-Dihydroorotic acid (D7128), Oil Red O (O0625), periodic acid (P7875), PEG 400 (91893), potassium chloride (P9541), rotenone (R8875), sodium chloride (S9888), sodium lactate (L7022), sodium phosphate monobasic (RDD007), sodium pyruvate (P2256), sucrose (S0389), N,N,N',N'-Tetramethyl-p-phenylenediamine [TMPD (T7394)] were purchased from Sigma-Aldrich. HPLC-grade acetonitrile (A955), formic acid (A117), methanol (A456), and water (W6) were purchased from Fisher Scientific. Eosin Y (SE23) and Tris base (BP301) was purchased from Fisher Scientific. GSH (G597951), GSSG (G597970), $^{13}$C$_2$;$^{15}$N GSH (G597952), and $^{13}$C$_4$;$^{15}$N$_2$ GSSG (G597972) were purchased from Toronto Research Chemicals. [U-$^{13}$C] lactate (CLM-1579), [U-$^{13}$C] pyruvate (CLM-2440), [U-$^{13}$C] glucose (CLM-1396) were purchased from Cambridge Isotope Laboratories. Schiff's Reagent was purchased from Electron Microscopy Sciences (26052–06). pAAV.TBG.PI.eGFP.WPRE.bGH (AAV-GFP) and pAAV.TBG.PI.Cre.rBG (AAV-Cre) were a gift from James M. Wilson (Addgene viral prep #105535-AAV8 and #107787-AAV8). Antibodies used: Tom20 (Proteintech, 11802), Pyruvate carboxylase (Proteintech, 16588), Actin (Proteintech, 66009), Ndufa9 (ThermoScientific, 459100), β2-microgobulin (ThermoScientific, 701250), Cox10 (Abcam, ab84053).

## Mice

The *Cox10*$^{f/f}$ mice (strain 024697) were purchased from The Jackson Laboratory and were maintained on a C57BL6 background. The *Ndufa9*$^{f/f}$ mice were obtained from David McFadden (UT Southwestern) and were on a C57BL6 background; generation of these mice are described here (*Wang et al., 2022*). All mice were housed in the Animal Resource Center at University of Texas Southwestern Medical Center under a 12 hr light-dark cycle and fed ad libitum. Prior to all experiments, mice were fasted for 16–18 hr over night (unless otherwise indicated). All mouse experiments were performed according to protocols approved by the Institutional Animal Care and Use Committee (IACUC) at University of Texas Southwestern Medical Center. Both male and female mice were used in all experiments; if sex differences were not present, both male and female mice were analyzed together. For genotyping, the following primers were used:

> *Cox10* Forward: GAGAGGAGTCAAGGGGACCT
> *Cox10* Reverse: GGCCTGCAGCTCAAAGTGTA
> *Ndufa9* Forward: TTGATTGCCTGTGAGCTTTG
> *Ndufa9* Reverse: TGCTAGGAAAGAGGCAGGTC

For AAV injections, virus was prepared in dilution buffer [NaCl (136 mM), KCl (5 mM), MgCl2 (1 mM), Tris (10 mM, pH 7.4), and glycerol (10%)] and injected into the retroorbital vein at 5x10$^{10}$ GC per mouse.

## Mitochondrial ETC activity measurements

Oxygen consumption rates (OCR) were measured using a Seahorse XFe96 Analyzer (Agilent Technologies) based on a previously published protocol (*Rogers et al., 2011*). Liver tissue was homogenized with 40 strokes of a dounce homogenizer in mitochondrial isolation buffer [HEPES (5 mM), sucrose (70 mM), mannitol (220 mM), MgCl2 (5 mM), KH2PO4 (10 mM), and EGTA (1 mM), pH 7.2]. Mitochondria were isolated via differential centrifugation: First nuclei and cell debris were removed by isolating supernatant after five spins at 600 g; and enriched mitochondria were then pelleted with a 10,000 g spin. Five µg of mitochondria were plated in an XFe96 well plate on ice, spun at 4700 RPM (Beckman Coulter S6096) for 2 min at 4 °C. Mitochondria were then supplemented with media containing isolation buffer with complex IV substrates to probe respiration. At the times indicated, ADP (final concentration 4 mM), oligomycin (2 µM), CCCP (2 µM), and either antimycin A (4 µM) or sodium azide (40 mM) was injected.

| No substrate | Isolation buffer (IB) |
| --- | --- |
| Complex IV | IB with ascorbate (10 mM)+TMPD (100 uM)+antimycin A (2 uM) |

For mouse embryonic fibroblast experiments, *Ndufa9$^{f/f}$* and *Ndufa9$^{-/-}$* MEFS were obtained as previously described (*Wang et al., 2022*). Briefly, E13.5 embryos were collected from *Ndufa9$^{f/f}$* pregnant females, minced and digested with trypsin for 45 min at 37 °C. Digested embryos were cultured in DMEM (Sigma-Aldrich, D6429) supplemented with 10%FBS, 1% penicillin/streptomycin, and immortalized with a lentiCRISPRv2 lentivirus expressing sgRNA targeting Trp53. Immortalized MEFs were subsequently transduced with Ad5-CMV-EGFP or Ad5-CMV-CreEGP adenovirus (University of Iowa Viral Vector Core), followed by culturing in the above media supplemented with 1 mM sodium pyruvate and 100 µg/ml uridine. GFP + cells were sorted by flow-cytometry to obtain purified *Ndufa9$^{f/f}$* and *Ndufa9$^{-/-}$* cells, which were then verified by genotyping and western blot analysis, as previously reported (*Wang et al., 2022*). For oxygen consumption measurements, *Ndufa9$^{f/f}$* and *Ndufa9$^{-/-}$* cells were plated overnight and exchanged into Seahorse assay buffer and measurements started. Mitochondrial oxygen consumption rates (mito-OCR) were calculated by subtracting non-mitochondrial respiration from baseline respiration.

Complex I activity measurements were performed using an immunocapture absorbance assay (ab109721; Abcam) following manufacturer's instructions. Approximately 100 mg of tissue were isolated from snap frozen mouse livers that had been stored at −80 °C. Tissue was homogenized in 500 µl PBS and protein concentrations were quantitated (DC Protein assay, Biorad). Samples were diluted with PBS to a concentration of 5.5 mg/ml and 90 µl were mixed with 10 µl of detergent (Abcam) and incubated on ice for 30 min. Samples were then spun down at 17,000 x g and 4 °C and the supernatants were collected and diluted to 250 µg/mL of protein. A total of 50 µg of each sample were loaded in duplicate onto the immunocapture plate and incubated for 3 hr. Experimental wells were then washed three times before adding 200 µL of reaction mixture. The OD450 was monitored over 30 min (SpectraMax), and the resulting reaction curves were fit by linear regression (Prism Graphpad) and slopes reflecting Complex I activity are reported by as mOD 450 / µg protein/min.

ATP measurements were carried out with an ATP Colorimetric/Fluorometric assay kit (K-354, BioVision Inc). Approximately 10 mg of tissue were isolated from snap frozen mouse livers that had been stored at −80 °C. Tissues were homogenized in 100 µl of ATP assay buffer and were rapidly deproteinized by PCA precipitation (K-808, BioVision Inc). Ten µl of deproteinized sample were diluted with 40 µl of ATP assay buffer and loaded in duplicate into a clear bottom 96 well plate. A total of 50 µl of developer solution was added to each well and the plate was incubated for 30 min at room temperature protected from light. The endpoint OD570 of sample wells was measured on a Spectramax iD3 spectrometer (SpectraMax, Molecular Devices) and the total amount of ATP was determined from a standard curve constructed with 0–10 nmol ATP. Reported values are normalized by tissue weight.

## qRT-PCR and qPCR measurements

Genomic DNA was isolated by ethanol precipitation and RNA was isolated from tissue using RNeasy Mini kits (Qiagen). Samples were run using the Luna Universal One-Step RT-qPCR Kit (New England Biolabs) on a CFX384 (Bio-Rad).

| Target | Primer Sequence |
|---|---|
| *Cox10* | F: AGGGTCAGCATCACCAATAC |
| | R: GGAGACACTTACCAGCATCAA |
| *Ndufa9* | F: GGAGACACTTACCAGCATCAA |
| | R: CCTCCTTTCCCGTGAGGTA |
| mtDNA (*ND2* mtDNA) | F: CCTATCACCCTTGCCATCAT |
| | R: GAGGCTGTTGCTTGTGTGAC |
| nDNA (*Pecam1*) | F: ATGGAAAGCCTGCCATCATG |
| | R: TCCTTGTTGTTCAGCATCAC |

## Histology of liver tissue

Fresh tissue was fixed overnight in 10% neutral buffered formalin, rinsed with 30% sucrose in PBS, and paraffin embedded. Ten micron sections were cut and placed on slides at –20 °C until staining. Sections were stained by either haemotoxylin and eosin, Oil Red O, or periodic acid Schiff base (PAS) and visualized under ×20 magnification on an Olympus IX83 microscope and processed with ImageJ (NIH).

## Plasma analysis

Blood was collected on ice with heparin and centrifuged at 5000xg for 10 min to obtain plasma. Plasma levels of alanine aminotransferase, albumin, aspartate aminotransferase, glucose, lactate, total bilirubin, and total protein were measured using a VITROS 250 Microslide at the UT Southwestern Metabolic Phenotyping Core.

## GSH/GSSG and NAD(P)(H) quantitation

For glutathione quantitation, 20 mg of frozen tissue was treated with 300 µL of 80% methanol with 0.1% formic acid and homogenized on ice. The samples were freeze-thawed, centrifuged to remove protein, and the supernatant was dried. Dried samples were reconstituted in 1 mL water with 0.1% formic acid, diluted, spiked with internal labeled standards, and analyzed as previously described (*Ubellacker et al., 2020*; *Tasdogan et al., 2020*).

NAD(P)+/NAD(P)H was measured as previously described (*Lu et al., 2018*). Briefly, ~20 mg of tissue was homogenized on ice in cold 40:40:20 (acetonitrile: methanol: water) with 0.1 M formic acid buffer. Tissue was vortexed, chilled, quenched with 15% ammonium bicarbonate, and centrifuged to pellet protein. Supernatant was diluted in 10 mM ammonium bicarbonate and $^{15}N_5$-AMP was added as an internal standard and injected onto a Sciex 6500 Qtrap using a reverse-phase ion pairing method. Pelleted protein was lysed and quantified using the DC protein assay (Bio-Rad).

For MEFs, cells were plated in six-well dishes and allowed to adhere overnight; the following day cells were at ~80% confluence. Media was removed, cells were rinsed with ice cold saline, and metabolites were collected by scraping cells in dry ice-cold 80% methanol / 20% water with an internal standard (series of 13 C, 15N-labeled amino acids). Metabolite extracts were stored at –80 °C overnight and centrifuged the following day to remove debris. Raw AUC values were obtained using MultiQuant and TIC values for normalization were obtained using Analyst (SCIEX software for MS analysis).

## Proteomics analysis of liver tissue

Protein homogenates were prepared from either whole liver tissues or enriched mitochondria. For whole liver tissue, approximately 100 mg of tissue were isolated from snap frozen mouse livers that had been stored at –80 °C. Tissue was homogenized in 500 µl PBS supplemented with protease inhibitors (Thermo Fisher, 78425) and protein concentrations were quantitated (DC Protein assay, Biorad). A

total of 50 µg of each sample was solubilized in 1% SDS, 50 mM Tris pH 8.0 at a concentration of 5 mg/ml, run on a 4–20% Mini-PROTEAN TGX precast protein gel (BioRad) into the top of the resolving portion of the gel, stained with Coomassie Blue, and destained. For enriched mitochondria, approximately 100 mg of liver tissue was homogenized with 40 strokes of a dounce homogenizer in mitochondrial isolation buffer [HEPES (5 mM), sucrose (70 mM), mannitol (220 mM), MgCl2 (5 mM), KH2PO4 (10 mM), and EGTA (1 mM), pH 7.2] supplemented with protease inhibitors (ThermoFisher, 78425). Mitochondria were isolated via differential centrifugation: First nuclei and cell debris were removed by isolating supernatant after five spins at 600 g; and enriched mitochondria were then pelleted with a 10,000 g spin. Protein concentrations were quantitated (DC protein assay, BioRad), and 50 µg of each sample was solubilized in 1% SDS, 50 mM Tris pH 8.0 at a concentration of 5 mg/ml. Samples were run on a 4–20% Mini-PROTEAN TGX precast protein gel (BioRad) into the top of the resolving portion of the gel, stained with Coomassie Blue, and destained.

Samples were analyzed at the UT Southwestern Proteomics Core. Gel slices were cut into 1 mm³ cubes, and digested overnight with trypsin (Pierce 90058) following reduction and alkylation with DTT and iodoacetamide (Sigma–Aldrich I6125). The samples then underwent solid-phase extraction cleanup with an Oasis HLB plate (Waters) and the resulting samples were injected onto an Orbitrap Fusion Lumos mass spectrometer coupled to an Ultimate 3000 RSLC-Nano liquid chromatography system. Samples were injected onto a 75 µm i.d., 75 cm long EasySpray column (Thermo) and eluted with a gradient from 0–28% buffer B over 90 min. Buffer A contained 2% (v/v) ACN and 0.1% formic acid in water, and buffer B contained 80% (v/v) ACN, 10% (v/v) trifluoroethanol, and 0.1% formic acid in water. The mass spectrometer operated in positive ion mode with a source voltage of 1.8 kV, ion transfer tube temperature of 275 °C, MS1 AGC target of 400000, MS1 maximum injection time of 50ms, intensity threshold of 5000, MS2 AGC target of 10000, MS2 maximum injection time of 35ms, MS2 isolation window of 1.6 Da. MS1 scans were acquired at 120,000 resolution in the Orbitrap and up to 10 MS/MS spectra were obtained in the ion trap for each full spectrum acquired using higher-energy collisional dissociation (HCD) for ions with charges 2–7. Dynamic exclusion was set for 25 s after an ion was selected for fragmentation.

Raw MS data files were analyzed using Proteome Discoverer v2.4 SP1 (Thermo), with peptide identification performed using Sequest HT searching against the mouse protein database from UniProt (downloaded January 2022; 17,062 sequences) assuming a trypsin digestion (cleavage after Lys and Arg except when immediately followed by Pro). Fragment and precursor tolerances of 10 ppm and 0.6 Da were specified, and three missed cleavages were allowed. Carbamidomethylation of Cys was set as a fixed modification, with oxidation of Met set as a variable modification. The false-discovery rate (FDR) cutoff was 1% for all peptides and all PSMs were validated with the Percoloator node within Proteome Discoverer. PSMs found in only a subset of samples were re-searched to identify peptides based on retention time and mass. Protein abundance was quantitated based on the total ion count for all identified peptides in each sample and only proteins notated with FDR <1% were considered. Protein abundances were normalized after log2 transform according to previously published protocols using Microsoft Excel 2019 (*Aguilan et al., 2020*): First, abundance values were log2 transformed, then each sample was normalized by the average abundance and data distribution width. Missing values were imputed as random values centered around 2.5 standard deviations below the mean abundance value. For samples with >4 replicates, Shapiro-Wilk tests were performed to test for normal distributions. If samples were not normally distributed, Mann-Whitney tests were used to calculate p-values. For normally distributed data, F-tests were used to assess equal variance, followed by two-tailed homoscedastic (or heteroscedastic) tests to calculate p-values. Unsupervised hierarchical clustering was performed on z-score transformed values in MATLAB (Mathworks, Inc). Mitochondrial-localized proteins were classified based on their presence in MitoCarta3.0 (*Calvo et al., 2016*). Significantly up and downregulated proteins were identified based on log$_2$(fold change)>1 or<1 (respectively) and p-value <0.05. Gene ontology analysis on up or downregulated proteins was performed using DAVID (https://david-d.ncifcrf.gov/home.jsp); for each analysis, the complete list of detected proteins was used as a background, and pathways from Biological Processes, Cellular Components and Molecular Function (All and Direct) were assessed. Pathways with a FDR <0.05 are reported in *Supplementary file 2*. Gene set enrichment analysis was performed using GSEA software (*Subramanian et al., 2005*; *Mootha et al., 2003*) using the following gene-ontology pathways: GO:0006633 (Fatty Acid Synthesis),

GO:0019395 (Fatty Acid Oxidation), GO:0008206 (Bile Acid Metabolic Process), GO:0016491 (Oxidoreductase Activity).

## In vivo isotope tracing

All in vivo isotope tracing experiments were performed in conscious mice after overnight fasting, based on previously published protocols (*Faubert et al., 2021*; *Davidson et al., 2016*). On the morning of the experiment, catheters (Braintree Scientific MRE-KIT 025) were surgically implanted into the right external jugular vein of mice. For steady state euglycemic infusion of [U-$^{13}$C]glucose, a total dose of 8 g/kg body mass (dissolved in 1000 µl saline) was continuously infused at 2.5 µl min$^{-1}$ (total infusion time of 3 hr). At the end of the infusion, mice were euthanized and tissue was immediately harvested, snap frozen in liquid N2, and stored at –80 °C prior to GC-MS analysis.

## Metabolomics analysis

Flash frozen liver tissue (20 mg) was pulverized with a mortar and pestle on liquid nitrogen (H37260-0100, Bel-Art Products), 80% methanol was added, and samples subjected to three freeze-thaw cycles. For plasma, 20 µL was extracted with 0.2 mL cold acetone. Protein was removed by centrifugation and supernatant dried (SpeedVac, ThermoFisher).

Data acquisition was performed by reverse-phase chromatography on a 1290 UHPLC liquid chromatography (LC) system interfaced to a high-resolution mass spectrometry (HRMS) 6550 iFunnel Q-TOF mass spectrometer (MS) (Agilent Technologies, CA). The MS was operated in both positive and negative (ESI +and ESI-) modes. Analytes were separated on an Acquity UPLC HSS T3 column (1.8 µm, 2.1x150 mm, Waters, MA). The column was kept at room temperature. Mobile phase A composition was 0.1% formic acid in water and mobile phase B composition was 0.1% formic acid in 100% ACN. The LC gradient was 0 min: 1% B; 5 min: 5% B; 15 min: 99% B; 23 min: 99% B; 24 min: 1% B; 25 min: 1% B. The flow rate was 250 µL min-1. The sample injection volume was 5 µL. ESI source conditions were set as follows: dry gas temperature 225 °C and flow 18 L min-1, fragmentor voltage 175 V, sheath gas temperature 350 °C and flow 12 L min-1, nozzle voltage 500 V, and capillary voltage +3500 V in positive mode and −3500 V in negative. The instrument was set to acquire over the full m/z range of 40–1,700 in both modes, with the MS acquisition rate of 1 spectrum s-1 in profile format. Raw data files were processed using Profinder B.08.00 SP3 software (Agilent Technologies, CA) with an in-house database containing retention time and accurate mass information on 600 standards from Mass Spectrometry Metabolite Library (IROA Technologies, MA) which was created under the same analysis conditions. The in-house database matching parameters were mass tolerance 10 ppm; retention time tolerance 0.5 min. Peak integration result was manually curated in Profinder for improved consistency. Metabolite abundances were normalized based on the total ion count for all identified metabolites. Unsupervised hierarchical clustering was performed on z-score transformed values in MATLAB (Mathworks, Inc), and adjusted p-values were calculated using multiple t-test with adjust for multiple comparisons in Graphpad Prism.

GC-MS analysis was used to quantitate α-hydroxybutyrate and α-ketobutyrate, as well as measure mass isotopomer distributions for the lactate/pyruvate tolerance test and [U-$^{13}$C]glucose infusions. Dried metabolites were derivatized to form methoxime-TBDMS adducts by incubating with 1% methoxyamine hydrochloride (Sigma-Aldrich) in pyridine at 70 °C for 15 min followed by addition of *N-tert*-Butyldimethylsiyl-*N*-methyltrifluoroacetamide (MTBSTFA, Sigma-Aldrich) for 1 hr. Derivatized samples were analyzed using an Agilent Technologies 7890B gas chromatographer with a HP-5MS 5% phenyl methyl Silox column (Agilent) coupled to an Agilent Technologies 5977 A mass spectrometer. The observed distributions of mass isotopologues were corrected for natural abundance (*Fernandez et al., 1996*).

## Lactate/pyruvate tolerance test

Lactate/Pyruvate tolerance tests were performed as previously described (*Cappel et al., 2019*). Briefly, mice were fasted for 16 hr overnight and given an IP injection of a 1.5 mg/g of a 10:1 sodium lactate: sodium pyruvate solution. 40% of the sodium lactate and sodium pyruvate were [U-$^{13}$C] sodium lactate and [U-$^{13}$C] sodium pyruvate. Blood glucose was measured 0, 10, 20, and 30 min post injection using a blood glucometer (Contour Next) and lactate was measured using a lactate meter (Nova Biomedical

Lactate Plus). Fractional gluconeogenesis (GNG) was determined as previously described (**Kelleher, 1999**), using the following formula:

$$GNG = (M1_{glucose} + M2_{glucose} + M3_{glucose})/(2 * M0_{lactate} * (M1_{lactate} + M2_{lactate} + M3_{lactate}))$$

where Mx represents the fractional enrichment of the m+x species for the indicated metabolite.

## Magnetic resonance imaging experiments

Experiments were performed on mice at 4 weeks post-AAV administration. MR imaging was performed on a 7T pre-clinical scanner (Bruker Biospec, Germany) using a 40 mm Millipede radio frequency coil for transmitting and receiving. Mice were anesthetized with 1.5–2.5% Isoflurane. Mice breathed medical air (21% O2) delivered via a nose cone, in the baseline acquisition and then the gas was switched to 100% O2 (Airgas, Radnor Township, PA, USA). The gas flow rate was set to 400 ml/min. A pneumonic sensor placed on the abdomen was used for respiratory monitoring and prospective triggering (SA Instruments, Stony Brook, NY). The animal's ambient temperature was maintained at 28 °C using an MR Compatible Small Rodent Air Heater System (SA Instruments, Stony Brook, NY). For a given animal, the breathing rate was maintained at 40 ± 5 bpm by varying the Isoflurane levels by ± 0.5%. At each breathing challenge, images were acquired to generate T1 and T2* maps of the liver in the coronal plane.

### T1 method

A respiratory triggered, radio frequency spoiled, flow compensated cine gradient echo method (FLASH) was used. To suppress the motion artifacts from cardiac motion, a saturation pulse was applied to the thoracic region. Two cines (number of frames =65) were acquired with two different flip angles ($\alpha$=5° and 20°). A steady state signal was achieved after 55 frames. Frames 55–65 were averaged to improve the signal-to-noise ratio. Pixel-by-pixel apparent (uncorrected for B1 inhomogeneity) T1 maps were calculated using the averaged signal intensities (S) according to **Helms et al., 2008**, $S\alpha = -2\frac{TR}{T1}\left(\frac{S}{\alpha}\right) + const$. Cine FLASH parameters were: TR/TE =10ms/2.3ms, FOV =40 x 40 mm$^2$, matrix =128 x 128, NEX =1, slice thickness =1 mm, number of segments =1, number of frames =65, in-plane resolution =0.3 mm x 0.3 mm.

Using the apparent T1 maps, and noting that any spatial dependencies due to B1 inhomogeneity are canceled out in relative measures, a map of the actual T1 difference between the gas challenges (Air vs. 100 %O2) was calculated as: $\Delta T1 = 100 * (T1_{O2} - T1_{air})/T1_{air}$. The mean relative $\Delta T1$ of the liver was calculated in a manually segmented liver region. Pixels representing blood vessels were automatically excluded from this region by thresholding to mean of liver T1 tissue ±2 × standard deviations. The mean of liver T1 was calculated from a region of interest that did not include blood vessels.

### T2* method

A respiratory triggered, bipolar-multi gradient echo (MGE) sequence was used. Sixteen echoes from positive gradients were acquired at 2.5ms intervals. The scan plane geometry and the orientation was identical to that of the T1 method. MGE parameters were: TR = 300ms, TE = 2.5 ms – 47.5 ms, NEX = 4. T2* maps were generated by non-linear least square fitting the echo signals (s) to an exponential model: $A * e^{-B*TE} + C$. The mean relative $\Delta T2^*$ was calculated as similar to $\Delta T1$.

### Perfusion method

FAIR arterial spin labeling technique was used to calculate the liver perfusion in a single-slice in the axial plane during the session where mice breathed medical air (21% O2). The FAIR sequence was a multi-echo spin-echo with a 13.6ms inversion pulse. A slice select inversion with an inversion slab of 4 mm was followed by a global inversion. The inversion time was set to 2000ms. A reference image was acquired without the inversion pulse. Other imaging parameter were: TR = 10 s, FOV = 30 x 25 mm$^2$, matrix = 128 x 64, NEX = 1, slice thickness = 1 mm, echo train length = 8. A pixel wise perfusion map was constructed as described previously (**Kim and Tsekos, 1997**).

## Statistical analyses

All data represent mean and standard deviations from biological replicates. No statistical tests were used to predetermine sample size. Animals were randomly allocated into experimental groups. No blinding or masking of samples was performed. Data sets for each group of measurement was tested for normality using the Shapiro-Wilk test. If the data was not normally distributed, the data was log-transformed and retested for normality. For normally-distributed data, groups were compared using the two-tailed Student's t-test (for two groups), or one-way ANOVA or two-way ANOVA (>2 groups), followed by Tukey's or Dunnett's test for multiple comparisons. For data that was not normally distributed, we used non-parametric testing (Mann-Whitney or Kolmogorov-Smirnov tests for two groups and Kruskal-Wallis test for multiple groups), followed by Dunn's multiple comparisons adjustment. For metabolomics and proteomics data sets, abundances were compared by multiple t-tests, followed by Bonferroni correction of p-values. Proteomics and metabolomics analysis were performed once using biological replicates (individual mice); for other experiments, multiple (two to four) independent experiments with biological replicates (individual mice) were performed for reported data, and the number of biological replicates are indicated in the figures. No data were excluded.

## Acknowledgements

We thank the Hao Zhu lab for assistance with liver experiments, the UT Southwestern Metabolic Phenotyping Core for plasma analysis, the UT Southwestern Proteomics Core, the CRI Metabolomics Facility, the Biochemistry Metabolomics Core, the UT Southwestern Small Animal Imaging Resource for help with various experiments and members of the Mishra and DeBerardinis laboratories for helpful discussions and suggestions during this project. Diagrams were created with BioRender.com. Funding: This work was supported by funding from the United Mitochondrial Disease Foundation (Research Grant to PM), the National Institutes of Health (1DP2ES030449-01 from NIEHS to PM, 1R01AR073217-01 from NIAMS to PM, 1F31-DK122676 from NIDDK to NPL), the Moody Medical Research Institute (Research Grant to P.M.), and the National Science Foundation (GRFP 2019281210 award to SDS).

## Additional information

### Competing interests

Ralph J DeBerardinis: is an advisor to Agios Pharmaceuticals. The other authors declare that no competing interests exist.

### Funding

| Funder | Grant reference number | Author |
| --- | --- | --- |
| United Mitochondrial Disease Foundation | | Prashant Mishra |
| National Institutes of Health | 1DP2ES030449-01 | Prashant Mishra |
| National Institutes of Health | 1R01AR073217-01 | Prashant Mishra |
| National Institutes of Health | 1F31-DK122676 | Nicholas P Lesner |
| Moody Medical Research Institute | | Prashant Mishra |
| National Science Foundation | GRFP 2019281210 | Spencer D Shelton |

The funders had no role in study design, data collection and interpretation, or the decision to submit the work for publication.

## Author contributions

Nicholas P Lesner, Conceptualization, Data curation, Formal analysis, Validation, Investigation, Visualization, Methodology, Writing - original draft, Writing – review and editing; Xun Wang, Zhenkang Chen, Anderson Frank, Cameron J Menezes, Sara House, Spencer D Shelton, Investigation; Andrew Lemoff, Resources, Formal analysis, Writing – review and editing; David G McFadden, Resources; Janaka Wansapura, Resources, Data curation, Formal analysis, Investigation, Visualization, Methodology, Writing – review and editing; Ralph J DeBerardinis, Resources, Writing – review and editing; Prashant Mishra, Conceptualization, Resources, Data curation, Formal analysis, Supervision, Funding acquisition, Visualization, Methodology, Writing - original draft, Writing – review and editing

## Author ORCIDs

Nicholas P Lesner (iD) http://orcid.org/0000-0001-9734-8828
Zhenkang Chen (iD) http://orcid.org/0000-0002-7919-5546
Cameron J Menezes (iD) http://orcid.org/0000-0001-5759-8099
Spencer D Shelton (iD) http://orcid.org/0000-0003-1236-5317
Andrew Lemoff (iD) http://orcid.org/0000-0002-4943-0170
Prashant Mishra (iD) http://orcid.org/0000-0003-2223-1742

## Ethics

All mouse experiments were performed according to protocols approved by the Institutional Animal Care and Use Committee (IACUC) at University of Texas Southwestern Medical Center (protocol 102654).

## Decision letter and Author response

Decision letter https://doi.org/10.7554/eLife.80919.sa1
Author response https://doi.org/10.7554/eLife.80919.sa2

# Additional files

## Supplementary files

• Supplementary file 1. Excel spreadsheet containing raw data and statistical analysis of proteomics and metabolomics data for *Ndufa9*$^{-/-}$ vs. *Ndufa9*$^{f/f}$ livers at 4 weeks post-AAV administration.

• Supplementary file 2. Excel spreadsheet containing raw data and statistical analysis of proteomics and metabolomics data for *Cox10*$^{-/-}$ vs. *Cox10*$^{f/f}$ livers at 8 weeks post-AAV administration.

• Supplementary file 3. Excel spreadsheet containing raw data for panels presented in the figures and figure supplements.

• Transparent reporting form

## Data availability

Proteomics datasets have been deposited into the PRIDE database (identifier PXD031685), and metabolomics datasets have been deposited into Metabolomics Workbench (identifier PR001484). All other data are provided within the article and supplementary files.

The following datasets were generated:

| Author(s) | Year | Dataset title | Dataset URL | Database and Identifier |
|---|---|---|---|---|
| Lesner NP, Mishra P | 2022 | Mitochondrial Hepatopathy Proteomics – Whole Tissue | https://doi.org/10.6019/PXD031685 | PRIDE, 10.6019/PXD031685 |
| Lesner NP, Mishra P | 2022 | Differential requirements for mitochondrial electron transport chain components in the adult murine liver | https://doi.org/10.21228/M8WT54 | Metabolomics Workbench, 10.21228/M8WT54 |

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
