## [Editor Report]

The commonly accepted view is that complex I of the mitochondrial respiratory chain is required for oxidative phosphorylation. This is an important paper, which discovers that complex I is not required for maintaining metabolically functional mitochondria in the liver, explaining a lack of liver defects in complex I deficiency patients. The study includes compelling data of proteomics and metabolomic tracer analyses representing a resource for a broad community with interest in biochemistry and metabolism of the cell as well as biomedical research. Based on this data the authors propose an alternative source of electrons for the respiratory chain in the mouse liver – the concept of potentially large impact on our understanding of metabolism.

---

## [Decision Letter]

**Decision letter after peer review:**

[Editors’ note: the authors submitted for reconsideration following the decision after peer review. What follows is the decision letter after the first round of review.]

Thank you for submitting the paper "Differential requirements for mitochondrial electron transport chain components in the adult murine liver" for consideration at *eLife*. Your initial submission has been assessed by a Senior Editor in consultation with members of the Board of Reviewing Editors. Although the work is of interest, we regret to inform you that the findings at this stage are too preliminary for further consideration at *eLife*.

The reviewers find the topic and findings potentially interesting and significant. The reviewers have brought up serious criticism (comments appended below), which precludes invitation to revise at this stage. However, they encourage you to resubmit. and new manuscript to *eLife* with a more full picture including a mechanistic evidence requested in their comments.

Specifically, the evidence for the involvement of DHODH is not sufficient. Thus, this interesting and important conclusion remains not fully proven. Furthermore, the proteomic and metabolic analyses are of insufficient depth and/or not analysed thoroughly to its maximum potential.

*Reviewer #1 (Recommendations for the authors):*

In this work, Lesner and colleagues study the effects of mitochondrial electron transport chain dysfunction at the level of mitochondrial complex I in comparison to complex IV in mouse liver. They report no adverse effects on liver functions in a complex I-deficient mouse model, whereas complex IV deficiency results in decreased liver function. To provide further insight into the mechanisms underlying these different phenotypic outcomes, the authors collect experimental data for alterations in the redox status as well as the proteome and metabolome in complex I- and complex IV-deficient mouse liver. These data generally support the notion that complex I activity is dispensable for liver function. The authors explain this finding by providing evidence for the involvement of mitochondrial enzyme dihydroorotate dehydrogenase as an alternative electron donor to maintain respiratory chain activity. These data add to the field of mitochondrial diseases associated with the dysfunction of mitochondrial respiratory chain complexes and tissue-specific phenotypes.

The authors generally present their data in a clear and logical way, which makes the story easy to follow. However, the description of proteomics experiments including LC-MS analysis in Materials and methods is largely incomplete and needs to be fully revised to provide detailed information about sample preparation, protein identification and quantification. In addition, Supplemental Tables for proteomic (and metabolomics) data are insufficient lacking information about gene names, identified peptides per replicate, ratios, calculated p-values etc. Proteomics data should also be deposited to a public database (e.g. PRIDE).

The authors performed label-free proteomics to investigate changes in the proteome of ndufa9-/- and cox10-/- mice vs. control mice. It should be noted that only less than 1/3 of the proteome was covered and thus they likely missed also some further regulated proteins in their analysis. Since they focus on mitochondrial processes, it is suggested to also analyze mitochondria-enriched fractions. Furthermore, the authors should improve their data presentation. For example, it would be informative to prepare additional plots showing the distribution of individual OXPHOS complexes in ndufa9-/- and cox10-/- liver samples. Do they generally shift to more positive or negative log2 values? In addition, the analysis of OXPHOS complexes by blue native PAGE is recommended to reveal changes in supercomplex formation or assembly states.

To provide further insight into compensatory effects in cox10-/- livers, GO term (BP, CC, MP) overrepresentation analysis of both downregulated and upregulated proteins should be performed. More specifically, the authors should provide information about the down-regulated proteins other than only complex IV components and upregulated proteins other than mitochondrial components. They should analyze the proteomic data in more detail, including the annotation of upregulated non-mitochondrial and mitochondrial proteins in Figures 2E and 4E. Furthermore, how do the authors explain the more than 2-fold increase in mitochondrial content in cox10-/- livers? It might be of interest to acquire EM or immunofluorescence images on these mouse liver samples.

Based on GO term enrichment analysis, the authors state that several metabolic pathways were enriched in cox10-/- livers. However, what is missing is a correlation between the observed changes in the proteome and metabolome. Do changes in the levels of specific metabolites result from the up- or down-regulation of proteins of the respective pathways?

In Figure 5B, quantification of Western blots should be performed and further replicates might likely be needed to make conclusions whether the levels of DHODH do not change in ndufa9-/- mice. In the reviewer's point of view, DHODH levels are slightly increased in liver from ndufa9-/- mice.

*Reviewer #2 (Recommendations for the authors):*

Strengths of the manuscript.

The fundamental data pillars supporting the authors top-line conclusion are solid – i.e. that liver damage is only apparent in complex IV, but not complex I, knockout adult mice.

After complex IV knockout the circulating clinical-biomarkers of liver injury (AST/ALT/Bilirubin) are elevated with statistical significance. In agreement, the complex IV knockout livers exhibits marked damage in the pathology assessment. Furthermore, the authors confirm robust changes in the liver proteome and metabolome following complex IV knockout; these effects are commensurate with mitochondrial dysfunction and adaptive stress signalling. Effects of complex I deletion on liver homeostasis are either minor or not detectable within the parameters assessed here; the difference between the effect of deleting complex IV versus complex I is clear and justified by data presented in the manuscript.

In the context of mitochondrial biology there is general lack of conditional knockout studies in adult mice. Accordingly, this study offers novel and unexpected insights into the relationship between respiratory chain components and hepatic function – the observation that adult mouse liver exhibit greater dependence on complex IV is of notable conceptual advance.

Weakness of the manuscript.

The key weakness of the manuscript is the lack of convincing mechanistic rationale for the differential effects of complex IV and complex I knockout on adult mouse liver.

The authors show that complex I knockout ~fully suppresses respiration (OCR) in freshly isolated hepatocytes (Figure 1C). Because the NADH/NAD+ ratio is unaffected in complex I-knockout livers the authors infer that another enzyme in the respiratory chain (dihydroorotate dehydrogenase (DHODH)) is catalysing NADH oxidation (to stop NADH accumulation/build-up) – however NADH oxidation by DHODH drives respiration (as depicted in Figure 5A) but Figure 1C shows very clearly that respiration rates are negligible in hepatocytes devoid of complex I activity/expression. Therefore, the authors explicitly show in their own data that respiratory activity is not maintained by DHODH in the absence of complex I.

Furthermore the authors do not present respiration/OCR data in hepatocytes isolated from complex IV knockout livers; there we cannot assess if hepatic respiration is differentially affected by complex I versus complex IV knockout- if the respiration rates are equivalently affected this would show that liver damage following complex IV deletion are not ascribable to low respiration rates/activity.

Next, the authors attempt to present data to support their model that DHODH is central to maintaining liver homeostasis in livers devoid of complex I, and treat mice with a small-molecule-inhibitor BRQ (an inhibitor of DHODH). The authors observe changes in the hepatic NADH/NAD+ ratio and interpret this data to support the DHODH model. However, the data is inadequate to support the authors conclusion;

1) There is already scant respiration in the complex I knockout hepatocytes (DHODH is not compensating for complex I deletion); 2) the authors do not show if BRQ effects respiration in control hepatocytes nor the remnant respiration in hepatocytes from complex I or complex IV livers; 3) whilst BRQ suppresses liver NADH /NAD+ its' impossible to ascribe this pharmacological effect to DHODH-specific inhibition ( it could be an alternative enzyme/target). The authors are not in a position to determine specificity of BRQ to DHODH and to make credible robust mechanistic interpretation; complementary genetic approaches would be required (i.e. showing DHODH knockout in the complex I knockout mouse liver recapitulates the complex IV knockout effect); 4) the authors present no in vivo evidence that BRQ treatment recapitulates the complex IV knockout phenotype – neither circulating markers (ALT/AST/bilirubin) nor pathology are presented.

Taken together the current model ascribing the lack of liver damage following complex I deletion to DHODH is not justified with existing data.

Recommendations:

1) For reasons outlined above the model ascribing the lack of liver damage following complex I deletion to DHODH is not justified with existing data. My recommendation is the data should be withdrawn from the manuscript – otherwise many months of experiments are required to support the possible role of DHODH. However, even without mechanistic insight into why complex IV deletion, but not complex I, exerts liver damage the finding in-itself is of sufficient novelty to justify publication.

2) Respiration/OCR data from the complex IV knockout hepatocytes is critical to aid data interpretation and provide mechanistic insight – it must be included and contrasted to the OCR in complex I knockout hepatocytes.

3) Whilst complex IV clearly exerts liver damage (~4x elevation in ALT and AST and histological staining) the level of damage (4-8 weeks after the knockout) is not overly severe and the animals do not undergo acute liver failure. The text and discussion need to better reflect this. Isn't the story here that mouse liver is not absolutely dependent on respiration to maintain function – otherwise we would expect rapid liver failure soon after the complex IV deletion?

4) Do the authors have data on hepatic ATP levels in absence of complex I and complex IV – both from liver tissue as well as the isolated hepatocytes? Are hepatic ATP levels maintained in both knockout models or is glycolysis maintaining ATP to ~homeostatic levels – the authors have shown respiration is essentially abolished upon complex I deletion, so how is ATP production supported? What is the ECAR in the complex I and complex IV knockout hepatocytes (the authors must have the ECAR data for complex I knockout because the OCR data is presented in figure 1). More generally the text doesn't address or discuss the concept of glycolytic adaptive capability in the absence of respiration.

5) In places the text is too assertive and not reflective of the data – more nuance and precision is required as to not over-interpret data, for example;

Page11 – 'The phenotype of cox10-/- livers noted above indicates that the mitochondrial ETC is required for liver homeostasis'.

How is this justified when the authors have already shown in figure 1 that the complex I knockout hepatocytes exhibit little respiration but liver homeostasis is not measurably effected? Furthermore, the authors do not show the respiration/OCR data in complex IV knockout hepatocytes.

'Together, these results indicate that complex I is largely dispensable for homeostatic function in the adult mouse liver, without significant metabolic or proteomic compensation'

Disagree. Respiration is dramatically reduced in complex I knockout hepatocytes – if no major adaption is ongoing how are hepatic ATP levels supported? There is presumably significant adaption to glycolysis and the PPP pathway to preserve ATP levels?

'Together, these results indicate that complex I is largely dispensable for homeostatic function in the adult mouse liver, without significant metabolic or proteomic compensation'

The authors have assessed liver health 4 and 8 weeks after complex I (and complex 4) deletion. The text needs to explicitly reflect these two sampling time points. Perhaps complex I knockout would affect hepatic function over a longer period?

Is 4 weeks knockout the shortest time point the authors have assessed? Maybe there is a transient stress/death response to complex I deletion (liver stress/death (ALT/AST rises)) but the liver is able to adapt to complex I loss and restore homeostasis before the initial 4 week sampling.

Therefore general statements regarding dispensability of complex I in liver need much greater refinement and nuance.

6) Complex I knockout livers (Supp figure 1D_ – 'Histological analysis of liver sections revealed no significant pathology, including no evidence of steatosis (Oil Red O staining) or glycogen loss (PAS staining)'.

They authors say 'significant', would they agree that are some changes in liver pathology (HandE and the Oil red O staining in particular) albeit they are far less severe compared to the cx4 knockout? If they agree the text needs to better reflect this.

7) Metabolomics for complex I knockout in figure 2 – is this data from 4 or 8 weeks? If its 4 weeks can the authors also show the 8 week data?

8) It will be helpful to label the charts for metabolites as' liver' or 'circulating' to make data interpretation clearer for reader.

*Reviewer #3 (Recommendations for the authors):*

Lesner and colleagues address the tissue specific phenotype seen in mitochondrial complex I deficiency, the most common cause of paediatric mitochondrial disease. Understanding this problem may lead to more personalised treatment strategies for patients with the disease. The authors leverage mouse models featuring liver specific knockout of complex I subunit Ndufa9 and complex IV assembly factor COX10, previously shown to result in loss of complex I and IV activity respectively. Using a range of classical biochemistry, pathology, proteomics and metabolomic approaches, the authors show that while loss of complex IV leads to severe impacts on cellular redox systems and metabolic compensation as expected, the most telling readout being NAD+/NADH ratio, complex I is not required to maintain metabolically normal mitochondria and livers from these animals are phenotypically similar to controls. Based on accumulation of metabolites in the pyrimidine synthesis pathway in complex IV deficient animals, the authors build a hypothesis that liver mitochondria preferentially utilise DHODH as a primary source of electrons for mitochondrial respiration. DHODH is a key enzyme in the pyrimidine synthesis pathway and an alternate source of electrons for the respiratory chain, effectively bypassing complex I by donating electrons directly to coenzyme Q which are then utilised by complex III (and subsequently passed to complex IV). Using the DHODH inhibitor brequinar the authors are able to generate a severe metabolic defect (the primary readout again NAD+/NADH ratio) in complex I deficient liver, but not controls. The main conclusion is that under normal conditions complex I is not the main entry point of electrons into the mitochondrial respiratory chain in liver, but rather pyrimidine synthesis, possibly supplemented by other sources. It is important to note that the main conclusion of preferential utilisation of pyrimidine synthesis derived electrons is drawn from a single experiment measuring the NAD+/NADH ratio. Additional metabolomic tracer studies using glutamine (and possibly other carbon sources aimed at delineating the other electron sources hinted at by the authors in the discussion) could be used to confirm if there is increased flux through pyrimidine synthesis.

The manuscript is well written, succinct and experiments are all of the highest quality. I have no major concerns in respect to the first conclusion that complex I deficient livers are phenotypically normal, and the resulting suggestion that complex I is not the major source of electrons for the ETC in liver. My concern is the conclusion that the main source is DHODH. This is drawn from a single (but apparently well controlled, though I'm not an expert on liver physiology) experiment using brequinar, with the NAD+/NADH ratio being the key readout. Increased flux doesn't always equate to changes in enzyme protein level or metabolite pool size, thus it not surprising that DHODH protein and metabolites in the pyrimidine synthesis pathway are not increased in abundance in ndufa9-/- livers. Indeed, except for dihydroorotate and orotate even in the extreme of BRQ treated animals you don't see accumulation of pyrimidine pathway metabolites. The authors should consider ways to measure flux directly, for example tracer studies using 13C-glutamine (or possibly H13CO3) could be used on primary hepatocytes from both knockouts. Inclusion of 13C-glucose would also go some way to addressing the possible increased flux through other pathways, e.g. complex II/TCA suggested by the authors in the discussion.

[Editors’ note: further revisions were suggested prior to acceptance, as described below.]

Thank you for resubmitting your work entitled "Differential requirements for mitochondrial electron transport chain components in the adult murine liver" for further consideration by *eLife*. Your revised article has been evaluated by Vivek Malhotra (Senior Editor) and a Reviewing Editor.

The reviewers have discussed their reviews with one another, and agree that the manuscript has significantly improved and is on track to publication after you complete a set of requested modifications to the text mostly for clarity of data presentation and its broad availability for the community. They also indicate the importance of one additional experiment, metaproteome of cox10-/- vs cox10f/f, which seems rather straightforward to provide unless rationally argued otherwise.

The Reviewing Editor has drafted this to help you prepare a revised submission.

Essential revisions:

1) In the revised manuscript the authors hypothesize that ETFDH is the major electron donor in the liver. This is mainly based on the elevation of related proteins and metabolites identified in proteomics and metabolomics data. The evidence for ETFDH is thus still circumstantial so conclusions should be cautious. Based on the tone of the authors in the main text I think they agree, however some of the statements could be softened in the abstract and discussion.

2) The authors should carefully describe the underpinnings of some of the less common techniques being used within the text narrative. Eg., this is the first time I've seen MR imaging and I felt I was just thrown into the deep end of what it measures. Likewise, the authors don't provide non-experts with sufficient background to understand tracer metabolomics. I also didn't find the schematic in Figure 3A particularly helpful as the use of dots to describe isotopologues doesn't really help the reader interpret the associated bar plots. This could probably be solved by a more detailed figure legend.

3) Additional points concerning metabolomics:

– Tracer metabolomics data should be included in the supplement.

– All metabolomics data should be made available in a public repository (e.g. Metabolomics workbench).

4) Data has been deposited in PRIDE as requested, however, the methods section in the manuscript requires some further work.

a) The authors wrote "50 ug per sample of protein homogenate from livers or enriched mitochondria (prepared as described above)", yet is it not clear which sample preparation steps described above they refer to. The authors should clearly state the steps they performed before they solubilized proteins in SDS/Tris.

b) The descriptions in the methods section still lack basic information.

Please add the following parameters to the LC-MS/MS data acquisition section:

AGC target for MS1 and MS2

Max IT for MS1 and MS2

Resolution for MS2

Isolation window for MS2

Intensity threshold

c) The authors wrote, "Fragment and precursor tolerances of 10 ppm and 0.6 Da were specified, and three missed cleavages were allowed". Please name the enzyme and state the cleavage rules for the search.

d) The authors wrote: "The false-discovery rate (FDR) cut-off was 1% for all peptides". Please provide the FDR cut-off for proteins as well. Please add information on whether PSMs were validated with Percolator (or another script) applied in the PD workflow.

e) There is clearly a mistake in the following sentence "Significantly up and downregulated proteins were identified based on log2(fold change) <1 or <1 (respectively) and p-value<0.05". Please correct it.

f) The authors wrote, "Protein abundance was quantitated based on the total ion count for all identified peptides in each sample, and normalized after log transform according to previously published protocols (60)". The authors should state in the methods section how the normalization was performed (rather than just referring to a paper).

g) The tables containing proteomics data look better, my additional suggestion would be to:

1) Add a unique peptide count per replicate (this parameter is more informative than peptide count).

2) Add a score parameter for protein identification.

5) Since the proteome coverage in the WCL samples was rather low (2.6-3k proteins, of which only 396-407 are annotated as mitochondrial in MitoCarta3.0), the authors were asked to analyze proteins in isolated mitochondrial fractions. The authors followed the recommendation only for ndufa9-/-, however, they did not do it for cox10-/-.

The additional analysis they performed yielded quantification data for 738 mitochondrial proteins in isolated mitochondria compared to 396 mitochondrial proteins in WCL samples, which clearly shows an improvement. As the proteomic analysis of isolated mitochondria ndufa9-/- vs ndufa9f/f in fact nearly doubled the number of quantified mitochondrial proteins reported, the authors should also include a similar analysis for the isolated mitochondria cox10-/- vs cox10f/f.

6) They were asked to perform GO term enrichment analysis for the cox10-/- vs cox10f/f WCL dataset. The authors performed the GO term enrichment analyses separately for mitochondrial and other than mitochondrial proteins, however, both mitochondrial and non-mitochondrial proteins may be associated with certain GO terms. The authors should add an additional term enrichment analysis for all the proteins upregulated or downregulated in the cox10-/- vs cox10f/f WCL dataset.

7) The authors were asked to provide a correlation between the observed changes in the proteome and metabolome in cox10-/- livers. Specifically, they were asked to examine whether changes in the levels of specific metabolites result from the up- or down-regulation of proteins of the respective pathways.

In response, the authors added Figure 5 —figure supplement 4 and included a short description in the manuscript. However, no specific examples were presented, e.g. up- or down-regulated level of enzyme A possibly contributed to up- or down-regulated level of metabolite X. Rather than (or in addition to) Figure 5 —figure supplement 4, another figure or a table linking specific enzymatic activities with metabolites should be included.

---

## [Author Response]

[Editors’ note: the authors resubmitted a revised version of the paper for consideration. What follows is the authors’ response to the first round of review.]

Reviewer #1 (Recommendations for the authors):In this work, Lesner and colleagues study the effects of mitochondrial electron transport chain dysfunction at the level of mitochondrial complex I in comparison to complex IV in mouse liver. They report no adverse effects on liver functions in a complex I-deficient mouse model, whereas complex IV deficiency results in decreased liver function. To provide further insight into the mechanisms underlying these different phenotypic outcomes, the authors collect experimental data for alterations in the redox status as well as the proteome and metabolome in complex I- and complex IV-deficient mouse liver. These data generally support the notion that complex I activity is dispensable for liver function. The authors explain this finding by providing evidence for the involvement of mitochondrial enzyme dihydroorotate dehydrogenase as an alternative electron donor to maintain respiratory chain activity. These data add to the field of mitochondrial diseases associated with the dysfunction of mitochondrial respiratory chain complexes and tissue-specific phenotypes.The authors generally present their data in a clear and logical way, which makes the story easy to follow. However, the description of proteomics experiments including LC-MS analysis in Materials and methods is largely incomplete and needs to be fully revised to provide detailed information about sample preparation, protein identification and quantification. In addition, Supplemental Tables for proteomic (and metabolomics) data are insufficient lacking information about gene names, identified peptides per replicate, ratios, calculated p-values etc. Proteomics data should also be deposited to a public database (e.g. PRIDE).

Proteomics data is now deposited into PRIDE (identified PXD031716; reviewer account details provided in manuscript). Supplemental Tables have been updated to include the requested information (gene names, peptides per replicate, fold change and p-values). The Methods section has been updated and includes details on sample preparation, details on LCMS instrumentation and peptide identification/quantitation.

The authors performed label-free proteomics to investigate changes in the proteome of ndufa9-/- and cox10-/- mice vs. control mice. It should be noted that only less than 1/3 of the proteome was covered and thus they likely missed also some further regulated proteins in their analysis. Since they focus on mitochondrial processes, it is suggested to also analyze mitochondria-enriched fractions. Furthermore, the authors should improve their data presentation. For example, it would be informative to prepare additional plots showing the distribution of individual OXPHOS complexes in ndufa9-/- and cox10-/- liver samples. Do they generally shift to more positive or negative log2 values? In addition, the analysis of OXPHOS complexes by blue native PAGE is recommended to reveal changes in supercomplex formation or assembly states.

We thank the reviewer for these suggestions, and have accordingly enhanced our analysis and discussion of the proteomics data. We agree that our analysis is not comprehensive of the complete proteome, and it possible some regulated proteins were missed. We have updated the text to discuss this caveat (lines 174-178). Our goal was not to complete characterize the proteome changes in response to complex I or IV loss, but to examine if the genetically modified livers exhibited signs of adaptation to loss of each mitochondrial ETC component.

As suggested, we performed proteomics on enriched mitochondrial samples in ndufa9^/-^ livers to potentially capture additional regulatory proteins; this analysis is presented in figure 2 —figure supplement 1, and Table S1. We identified and quantified 738 mitochondrial proteins, which represents ~64% of the mitochondrial proteome. Similar to our whole cell data, we do not observe a large number of changes, complex I components are downregulated, while other complexes are largely maintained (Figure 2 —figure supplement 1, Table S1).

As suggested, additional plots focusing on individual components of each OXPHOS complex are now provided in Figures 2H, Figure 2 —figure supplement 1B, and Figure 5H. In ndufa9^-/-^ livers, we observe isolated reduction of complex I components, with sparing of protein components from other complexes. In cox10^-/-^ livers, we observe isolated reduction of complex IV components, with a general increase in components from other complexes. Thus, it appears that each individual knockout allele specifically affects the abundance of components in the respective complex, without major declines in other complex components.

Thank you for the suggestion regarding BN-PAGE analysis. The relevance of supercomplex formation in C57BL6 mice is complicated: a mutation in the SCAF1 gene in this genetic background results in reduced assembly of multimeric cIV-containing supercomplexes (PMID 26928661, 23812712). As a result, it has been proposed that there are fewer supercomplexes and/or cIV-containing supercomplexes in this background. We’ve done some preliminary BN-PAGE analysis (Author response image 1) which is consistent with this interpretation: Loss of cox10 or ndufa9 results in loss of the individual respective complexes, loss of the I+III_2_+IV_1_ supercomplex; while cox10^-/-^ livers retain a I+III_2_+II_n_ supercomplex.

**Author response image 1. sa2fig1:** BN-PAGE analysis to analyze ETC complexes and supercomplexes from murine liver lysate. (A) BN-PAGE, followed by Coomassie staining, of isolated mitochondria extracted with 5g/g digitonin/protein ratio. 50mg of protein per lane were assessed by BN-PAGE, and bands are annotated based on previously published data (PMID 26928661). (B) Same as (A), except bands were visualized by in-gel complex I activity, which exhibits a purplish color. Bands with complex I activity are indicated. (C) Same as (A), excepts bands were visualized by in-gel complex IV activity, which exhibits a brown color. Bands with complex IV activity are indicated.

The proposed lack of multimeric cIV-containing supercomplexes in this genetic background is perhaps related to our central finding that cI is not required under homeostatic conditions. Specifically, by not participating in supercomplexes, cIV may be “freed” and therefore available to accept electrons from alternative donors (e.g., succinate, fatty acids/ETF, dihydroorotate, etc.). A detailed analysis of cIV-containing supercomplexes in this genetic background and others, combined with an analysis of the requirement for hepatic ndufa9 in the setting of modulating SCAF1 status would be required to fully address this question. We propose that this be addressed in future work. We have amended our discussion to address this caveat (lines 456-467).

To provide further insight into compensatory effects in cox10-/- livers, GO term (BP, CC, MP) overrepresentation analysis of both downregulated and upregulated proteins should be performed. More specifically, the authors should provide information about the down-regulated proteins other than only complex IV components and upregulated proteins other than mitochondrial components. They should analyze the proteomic data in more detail, including the annotation of upregulated non-mitochondrial and mitochondrial proteins in Figures 2E and 4E.

We have updated the manuscript to included additional analysis and discussion of our proteomic data according to these suggestions. All significant non-mitochondrial and mitochondrial proteins are annotated in Figure 2G; and we have annotated a number of nonmitochondrial and mitochondrial proteins in Figure 5G. For compensatory effects in cox10-/- livers, we have performed GO analysis as suggested, and the results are presented in Figure 5 —figure supplement 1,2, as well as Table S2. The results of these analyses are now discussed in the main text (lines 289-301, 307-313).

Furthermore, how do the authors explain the more than 2-fold increase in mitochondrial content in cox10-/- livers? It might be of interest to acquire EM or immunofluorescence images on these mouse liver samples.

Increases in mitochondrial content are common in both mouse models and patients with mitochondrial dysfunction (eg., PMID 17951359), and has been presumed to represent increased organellar biogenesis in response to the organellar defect. EM in cox10-deficient livers has been previously reported (PMID 17951359), which reports an accumulation of lipid droplets and mitochondria without apparent changes in organelle morphology. We have included immunofluorescence analysis by staining for the mitochondrial protein Tomm20 (Figure 2 —figure supplement 2; Figure 5 —figure supplement 5) which did not reveal significant alterations in ndufa9-depeleted livers, but increased mitochondrial content in cox10-depleted livers.

Based on GO term enrichment analysis, the authors state that several metabolic pathways were enriched in cox10-/- livers. However, what is missing is a correlation between the observed changes in the proteome and metabolome. Do changes in the levels of specific metabolites result from the up- or down-regulation of proteins of the respective pathways?

Thank you for this suggestion; we have accordingly updated the manuscript with additional gene set enrichment analysis and discussion (Figure 5 —figure supplement 4). Specifically, we do observe proteomic changes in several genes relating to fatty acid oxidation, bile acid metabolism and electron transfer activity, although overall enrichment/depletion in these pathways does not rise to statistical significance.

In Figure 5B, quantification of Western blots should be performed and further replicates might likely be needed to make conclusions whether the levels of DHODH do not change in ndufa9-/- mice. In the reviewer's point of view, DHODH levels are slightly increased in liver from ndufa9-/- mice.

During revision, we have eliminated this figure from the manuscript; however the requested analysis of DHODH levels is presented in Author response image 2. We do observe increases in DHODH abundance by western blot and proteomics in response to loss of complex I or complex IV, suggesting that this pathway may be compensating for loss of ETC function. However, as discussed in the current manuscript, we do not observe significant alterations in DHODH metabolite levels, suggesting that DHODH activity is not a major electron source for the hepatic ETC.

**Author response image 2. sa2fig2:** Quantitation of DHODH abundance in ETC-deficient livers. (A) DHODH abundance was assessed by western blot in *ndufa9^f/f^* and *ndufa9^-/-^* livers, and was quantitated relative to b2microglobulin (as a loading control). (B) Relative DHODH abundance based on proteomic analysis from *ndufa9* and *cox10* deficient livers.

Reviewer #2 (Recommendations for the authors):Strengths of the manuscript.The fundamental data pillars supporting the authors top-line conclusion are solid – i.e. that liver damage is only apparent in complex IV, but not complex I, knockout adult mice.After complex IV knockout the circulating clinical-biomarkers of liver injury (AST/ALT/Bilirubin) are elevated with statistical significance. In agreement, the complex IV knockout livers exhibits marked damage in the pathology assessment. Furthermore, the authors confirm robust changes in the liver proteome and metabolome following complex IV knockout; these effects are commensurate with mitochondrial dysfunction and adaptive stress signalling. Effects of complex I deletion on liver homeostasis are either minor or not detectable within the parameters assessed here; the difference between the effect of deleting complex IV versus complex I is clear and justified by data presented in the manuscript.In the context of mitochondrial biology there is general lack of conditional knockout studies in adult mice. Accordingly, this study offers novel and unexpected insights into the relationship between respiratory chain components and hepatic function – the observation that adult mouse liver exhibit greater dependence on complex IV is of notable conceptual advance.Weakness of the manuscript.The key weakness of the manuscript is the lack of convincing mechanistic rationale for the differential effects of complex IV and complex I knockout on adult mouse liver.

We agree that our manuscript does not completely reconcile the differential effects between complex I and complex IV, and have expanded our discussion to comment on this point. Complex I serves as a single potential entry point into the ETC, while complex IV is expected to accept electrons from all potential electron donors. Thus, our findings are most consistent with a model in which cI is not the major source of electrons in the hepatic ETC. As suggested below, we’ve simplified our analysis and discussion to reflect this, and removed data reflecting DHODH activity. At this point, our data suggests the hypothesis that ETFDH is a significant electron donor to the hepatic ETC. This is based on the buildup of glutarate and acyl-carnitines, as well as the similarities in phenotypes between animals/patients with cIV defects and animals/patients with ETFDH defects.

The authors show that complex I knockout ~fully suppresses respiration (OCR) in freshly isolated hepatocytes (Figure 1C). Because the NADH/NAD+ ratio is unaffected in complex I-knockout livers the authors infer that another enzyme in the respiratory chain (dihydroorotate dehydrogenase (DHODH)) is catalysing NADH oxidation (to stop NADH accumulation/build-up) – however NADH oxidation by DHODH drives respiration (as depicted in Figure 5A) but Figure 1C shows very clearly that respiration rates are negligible in hepatocytes devoid of complex I activity/expression. Therefore, the authors explicitly show in their own data that respiratory activity is not maintained by DHODH in the absence of complex I.

We apologize for the confusion. In our initial manuscript, the oxygen consumption measurements from in vitro cultured hepatocytes was meant as a corollary for measuring complex I function. We have now replaced this data with direct measurements of complex I activity in liver lysates (Figure 1C). The major conclusion is that removal of ndufa9 does remove complex I activity in vivo.

More generally, when hepatocytes (or most cell lines) are cultured in high glucose media (as in our previous Figure 1C), the observed respiration is primarily driven by complex I. Whether this represents the “in vivo” scenario is the major investigative point of this study. To avoid further confusion, we have removed all primary hepatocyte data from this manuscript and focuse on in vivo investigations.

Assessing hepatic respiratory function in vivo would allow us to assess the relevance of complex I or complex IV to oxygen consumption – however, O2 flux measurements in vivo are not developed. To address this, we have added new experiments and data in which we perform in vivo MR imaging in the livers of mice in order to assess oxygen status. Ndufa9-/- livers are indistinguishable from wild-type livers in these experiments (Figure 3E-K), indicating no significant deficit in oxygen consumption when complex I is removed. In contrast, we observe elevated oxygen levels in complex IV-deficient livers (Figure 6A-F), indicating that oxygen consumption is impaired in cox10-/- livers. Together, these data indicate that the mitochondrial ETC is required to maintaining oxygen levels in vivo; however, complex I is not required, suggesting that an alternative electron donor is feeding the mitochondrial ETC.

Furthermore the authors do not present respiration/OCR data in hepatocytes isolated from complex IV knockout livers; there we cannot assess if hepatic respiration is differentially affected by complex I versus complex IV knockout- if the respiration rates are equivalently affected this would show that liver damage following complex IV deletion are not ascribable to low respiration rates/activity.

See comments above for more detail. Briefly, we have removed cultured hepatocyte data to remove confusion between the in vitro and in vivo situation. We now provide an assessment of oxygen status in cox10-/- livers (Figure 6A-F), to assess the role of complex IV in oxygen consumption in vivo.

Next, the authors attempt to present data to support their model that DHODH is central to maintaining liver homeostasis in livers devoid of complex I, and treat mice with a small-molecule-inhibitor BRQ (an inhibitor of DHODH). The authors observe changes in the hepatic NADH/NAD+ ratio and interpret this data to support the DHODH model. However, the data is inadequate to support the authors conclusion;1) There is already scant respiration in the complex I knockout hepatocytes (DHODH is not compensating for complex I deletion); 2) the authors do not show if BRQ effects respiration in control hepatocytes nor the remnant respiration in hepatocytes from complex I or complex IV livers; 3) whilst BRQ suppresses liver NADH /NAD+ its' impossible to ascribe this pharmacological effect to DHODH-specific inhibition ( it could be an alternative enzyme/target). The authors are not in a position to determine specificity of BRQ to DHODH and to make credible robust mechanistic interpretation; complementary genetic approaches would be required (i.e. showing DHODH knockout in the complex I knockout mouse liver recapitulates the complex IV knockout effect); 4) the authors present no in vivo evidence that BRQ treatment recapitulates the complex IV knockout phenotype – neither circulating markers (ALT/AST/bilirubin) nor pathology are presented.Taken together the current model ascribing the lack of liver damage following complex I deletion to DHODH is not justified with existing data.

We completely agree with these comments, and as suggested below, we have revised the manuscript to remove the emphasis on DHODH.

Recommendations:1) For reasons outlined above the model ascribing the lack of liver damage following complex I deletion to DHODH is not justified with existing data. My recommendation is the data should be withdrawn from the manuscript – otherwise many months of experiments are required to support the possible role of DHODH. However, even without mechanistic insight into why complex IV deletion, but not complex I, exerts liver damage the finding in-itself is of sufficient novelty to justify publication.

We agree, and have removed this data from the manuscript. We now provide a simplified interpretation based on analysis of substrates which accumulate in response to complex IV depletion. Briefly, we specifically observe accumulation of ETFDH-dependent substrates, leading us to hypothesize that ETFDH is a significant electron donor to hepatic ETC under homeostatic conditions.

2) Respiration/OCR data from the complex IV knockout hepatocytes is critical to aid data interpretation and provide mechanistic insight – it must be included and contrasted to the OCR in complex I knockout hepatocytes.

See comments above; we have removed isolated hepatocyte data to avoid confusion between in vitro and in vivo scenarios. We agree that assessment of respiration in cIV-deficient livers is critical, and provide an in vivo assessment in Figure 6A-F.

3) Whilst complex IV clearly exerts liver damage (~4x elevation in ALT and AST and histological staining) the level of damage (4-8 weeks after the knockout) is not overly severe and the animals do not undergo acute liver failure. The text and discussion need to better reflect this. Isn't the story here that mouse liver is not absolutely dependent on respiration to maintain function – otherwise we would expect rapid liver failure soon after the complex IV deletion?

Thank you for this comment. We agree; it is surprising that these mice are not dependent on either complex I or complex IV for survival. We have amended our text to discuss this point (lines 429-431).

4) Do the authors have data on hepatic ATP levels in absence of complex I and complex IV – both from liver tissue as well as the isolated hepatocytes? Are hepatic ATP levels maintained in both knockout models or is glycolysis maintaining ATP to ~homeostatic levels – the authors have shown respiration is essentially abolished upon complex I deletion, so how is ATP production supported? What is the ECAR in the complex I and complex IV knockout hepatocytes (the authors must have the ECAR data for complex I knockout because the OCR data is presented in figure 1). More generally the text doesn't address or discuss the concept of glycolytic adaptive capability in the absence of respiration.

We have examined hepatic ATP levels, and find no significant differences in complex I or complex IV livers (provided in Figure 2A and Figure 5A), suggesting that the mitochondrial ETC is not absolutely required to maintain steady state energy levels. As discussed above, we have removed experiments from cultured hepatocytes to avoid confounding in vitro and in vivo data.

To assess changes in glycolytic capacity, we have performed [U-13C]glucose isotope tracing in conscious mice, which reveals that removal of neither ndufa9 nor cox10 significantly impact glucose handling in the liver (Figure 3A-D; Figure 3 —figure supplement 1; Figure 6G-I; Figure 6 —figure supplement 1). In addition, we do not observe upregulation of the glycolytic pathway in response to ndufa9 or cox10 removal based on gene ontology analysis.

5) In places the text is too assertive and not reflective of the data – more nuance and precision is required as to not over-interpret data, for example;Page11 – 'The phenotype of cox10-/- livers noted above indicates that the mitochondrial ETC is required for liver homeostasis'.How is this justified when the authors have already shown in figure 1 that the complex I knockout hepatocytes exhibit little respiration but liver homeostasis is not measurably effected? Furthermore, the authors do not show the respiration/OCR data in complex IV knockout hepatocytes.

See comments above for clarification with respect to hepatocyte data, and we have adjusted the text accordingly to point out caveats in our interpretation. Specifically, we do not observe that complex I is required for oxygen homeostasis in vivo, although it is clearly required under in vitro culturing conditions.

'Together, these results indicate that complex I is largely dispensable for homeostatic function in the adult mouse liver, without significant metabolic or proteomic compensation'Disagree. Respiration is dramatically reduced in complex I knockout hepatocytes – if no major adaption is ongoing how are hepatic ATP levels supported? There is presumably significant adaption to glycolysis and the PPP pathway to preserve ATP levels?

We apologize, as our previous in vitro data was clearly inconsistent with our in vivo interpretation. We agree that, in vitro, cI is clearly required for respiration.

In contrast to the in vitro scenario, our new data suggests that complex I is dispensable for oxygen homeostasis in vivo, suggesting that it is not significantly contributing to mitochondrial respiration. Based on our proteomic, metabolomic and isotope tracing analysis, we do not observe evidence of compensation, suggesting that complex I is dispensable in the adult murine liver. We have amended the text to discuss the caveats to our interpretation (lines 456-464).

'Together, these results indicate that complex I is largely dispensable for homeostatic function in the adult mouse liver, without significant metabolic or proteomic compensation'The authors have assessed liver health 4 and 8 weeks after complex I (and complex 4) deletion. The text needs to explicitly reflect these two sampling time points. Perhaps complex I knockout would affect hepatic function over a longer period?

We have modified the text and figure legends to clarify the sampling dates for these experiments. We now also present data from animals at 8 weeks post complex I removal (Figure 1 —figure supplement 3), which indicates no detectable deficits in hepatic function.

Is 4 weeks knockout the shortest time point the authors have assessed? Maybe there is a transient stress/death response to complex I deletion (liver stress/death (ALT/AST rises)) but the liver is able to adapt to complex I loss and restore homeostasis before the initial 4 week sampling.

Thank you for the suggestion; this is an interesting idea. We now present data from animals at an earlier timepoint (2 weeks after complex removal), which indicates no significant deficits in hepatic function (Figure 1 —figure supplement 2).

Therefore general statements regarding dispensability of complex I in liver need much greater refinement and nuance.6) Complex I knockout livers (Supp figure 1D_ – 'Histological analysis of liver sections revealed no significant pathology, including no evidence of steatosis (Oil Red O staining) or glycogen loss (PAS staining)'They authors say 'significant', would they agree that are some changes in liver pathology (HandE and the Oil red O staining in particular) albeit they are far less severe compared to the cx4 knockout? If they agree the text needs to better reflect this.

We provide improved quality images in our revised manuscript and blinded analysis has not revealed detectable differences in ndufa9-/- livers. We have amended the discussion to include these caveats, and agree that it is possible there are minor defects preset in complex I-knockout livers (lines 421-422).

7) Metabolomics for complex I knockout in figure 2 – is this data from 4 or 8 weeks? If its 4 weeks can the authors also show the 8 week data?

The metabolomics and proteomics data for ndufa9-/- livers was performed at 4 weeks post complex I removal, and we have amended the figure legends to indicate the sampling timepoints. Unfortunately, we do not have a similar datasets at 8 weeks post-AAV.

8) It will be helpful to label the charts for metabolites as' liver' or 'circulating' to make data interpretation clearer for reader.

We have included the appropriate labels for clarity in the figure panels and supplementary tables.

Reviewer #3 (Recommendations for the authors):Lesner and colleagues address the tissue specific phenotype seen in mitochondrial complex I deficiency, the most common cause of paediatric mitochondrial disease. Understanding this problem may lead to more personalised treatment strategies for patients with the disease. The authors leverage mouse models featuring liver specific knockout of complex I subunit Ndufa9 and complex IV assembly factor COX10, previously shown to result in loss of complex I and IV activity respectively. Using a range of classical biochemistry, pathology, proteomics and metabolomic approaches, the authors show that while loss of complex IV leads to severe impacts on cellular redox systems and metabolic compensation as expected, the most telling readout being NAD+/NADH ratio, complex I is not required to maintain metabolically normal mitochondria and livers from these animals are phenotypically similar to controls. Based on accumulation of metabolites in the pyrimidine synthesis pathway in complex IV deficient animals, the authors build a hypothesis that liver mitochondria preferentially utilise DHODH as a primary source of electrons for mitochondrial respiration. DHODH is a key enzyme in the pyrimidine synthesis pathway and an alternate source of electrons for the respiratory chain, effectively bypassing complex I by donating electrons directly to coenzyme Q which are then utilised by complex III (and subsequently passed to complex IV). Using the DHODH inhibitor brequinar the authors are able to generate a severe metabolic defect (the primary readout again NAD+/NADH ratio) in complex I deficient liver, but not controls. The main conclusion is that under normal conditions complex I is not the main entry point of electrons into the mitochondrial respiratory chain in liver, but rather pyrimidine synthesis, possibly supplemented by other sources. It is important to note that the main conclusion of preferential utilisation of pyrimidine synthesis derived electrons is drawn from a single experiment measuring the NAD+/NADH ratio. Additional metabolomic tracer studies using glutamine (and possibly other carbon sources aimed at delineating the other electron sources hinted at by the authors in the discussion) could be used to confirm if there is increased flux through pyrimidine synthesis.The manuscript is well written, succinct and experiments are all of the highest quality. I have no major concerns in respect to the first conclusion that complex I deficient livers are phenotypically normal, and the resulting suggestion that complex I is not the major source of electrons for the ETC in liver. My concern is the conclusion that the main source is DHODH. This is drawn from a single (but apparently well controlled, though I'm not an expert on liver physiology) experiment using brequinar, with the NAD+/NADH ratio being the key readout. Increased flux doesn't always equate to changes in enzyme protein level or metabolite pool size, thus it not surprising that DHODH protein and metabolites in the pyrimidine synthesis pathway are not increased in abundance in ndufa9-/- livers. Indeed, except for dihydroorotate and orotate even in the extreme of BRQ treated animals you don't see accumulation of pyrimidine pathway metabolites. The authors should consider ways to measure flux directly, for example tracer studies using 13C-glutamine (or possibly H13CO3) could be used on primary hepatocytes from both knockouts. Inclusion of 13C-glucose would also go some way to addressing the possible increased flux through other pathways, e.g. complex II/TCA suggested by the authors in the discussion.

We thank the reviewers for this comment and agree that our experiments do not indicate DHODH is the major electron source. Based on comments from all 3 reviewers, we have removed this data, and clarified our interpretation. In particular, we hypothesize that fatty acids and ETFDH may be a major electron source, via contributions from fatty acids and amino acids. This is because in complex IV-deficient livers, we do not observe a buildup of succinate or dihydo-orotate or glycerol-3-phosphate, and we do not observe changes in glucose handling based on [U-13C]glucose tracing (Figure 6G-K). However, we do observe a significant increase in long chain acyl-carnitine species, suggesting a significant defect in fatty acid oxidation. Consistent with this interpretation, we observe decreased α-hydroxybutyrate levels and significant increases in glutarate levels. Interestingly, mice and humans suffering from ETF deficiency exhibit similar metabolic, histological and functional deficits in the liver. Thus, we propose that ETF serves as a major electron source to the mitochondrial ETC, which alleviates a dependence on complex I. Future experiments to trace flux through ETF will be necessary to examine this model in depth, and we have amended our discussion to include this point (lines 437-443).

As discussed above, we have removed experiments with cultured hepatocytes to avoid confusion between the in vivo and in vitro scenarios. Briefly, we find that culturing hepatocytes in high glucose results in their dependence on complex I for respiratory function. To avoid this potential discrepancy between experimental setups, we have added data assessing oxygen consumption by MR imaging (Figure 3E-K, Figure 6A-F) in vivo, as well as in vivo [U-13C]glucose tracing (Figure 3B-D, Figure 6G-I).

[Editors’ note: what follows is the authors’ response to the second round of review.]

Essential revisions:1) In the revised manuscript the authors hypothesize that ETFDH is the major electron donor in the liver. This is mainly based on the elevation of related proteins and metabolites identified in proteomics and metabolomics data. The evidence for ETFDH is thus still circumstantial so conclusions should be cautious. Based on the tone of the authors in the main text I think they agree, however some of the statements could be softened in the abstract and discussion.

We agree that the role of ETFDH as the major electron donor is still circumstantial, as it is difficult to directly determine the flux of electrons into the ETC in an in vivo context. As suggested, we have modified the text in the abstract and discussion, please see lines 45-46 (sentence removed) and lines 525-526 (sentence removed). In the discussion, we have stated that our steady-state measurements do not directly inform on flux through fatty acid oxidation / ETFDH pathways, and future experiments will be necessary (lines 512-514).

2) The authors should carefully describe the underpinnings of some of the less common techniques being used within the text narrative. Eg., this is the first time I've seen MR imaging and I felt I was just thrown into the deep end of what it measures. Likewise, the authors don't provide non-experts with sufficient background to understand tracer metabolomics. I also didn't find the schematic in Figure 3A particularly helpful as the use of dots to describe isotopologues doesn't really help the reader interpret the associated bar plots. This could probably be solved by a more detailed figure legend.

As suggested, we have modified the text to include an additional description of the MR imaging experiments, geared towards non-experts; please see lines 229-249. In addition, we have updated our description of the tracer experiments to provide additional clarity for non-experts (lines 207-215).

We apologize for the confusion with respect to figure 3A. In our original figure, the colored (red) dots referred to the fate of heavy label atoms from [U-^13^C]glucose labeling. However, considering the significant amount of scrambling of labels that occurs in vivo, we chose to plot the total labeled fraction in Figure 3D, which sums the contributions from all individual isotopologues with at least one labeled carbon. We have modified the figure 3A to remove indicators of heavy labeled carbons, and it now simply summarizes some of the potential fates of glucose and lactate as they enter a hepatocyte.

3) Additional points concerning metabolomics:– Tracer metabolomics data should be included in the supplement.– All metabolomics data should be made available in a public repository (e.g. Metabolomics workbench).

We have updated the Supplementary files 1 and 2 to include tracer metabolomics data.

All metabolomics data has been deposited into Metabolomics workbench; accession codes are provided within the manuscript.

4) Data has been deposited in PRIDE as requested, however, the methods section in the manuscript requires some further work.a) The authors wrote "50 ug per sample of protein homogenate from livers or enriched mitochondria (prepared as described above)", yet is it not clear which sample preparation steps described above they refer to. The authors should clearly state the steps they performed before they solubilized proteins in SDS/Tris.

We apologize for the omission, and have now included a description of sample preparation (lines 9971015).

b) The descriptions in the methods section still lack basic information.Please add the following parameters to the LC-MS/MS data acquisition section:AGC target for MS1 and MS2Max IT for MS1 and MS2Resolution for MS2Isolation window for MS2Intensity threshold

We have now included these details in the methods section (lines 1025-1032). A resolution is not specified for MS2 as MS2 scans were obtained in the ion trap of the mass spectrometer where mass resolution is not set in order to allow faster scans and identification of more proteins.

c) The authors wrote, "Fragment and precursor tolerances of 10 ppm and 0.6 Da were specified, and three missed cleavages were allowed". Please name the enzyme and state the cleavage rules for the search.

Added (lines 1035-1036).

d) The authors wrote: "The false-discovery rate (FDR) cut-off was 1% for all peptides". Please provide the FDR cut-off for proteins as well. Please add information on whether PSMs were validated with Percolator (or another script) applied in the PD workflow.

Added (lines 1018-1022).

e) There is clearly a mistake in the following sentence "Significantly up and downregulated proteins were identified based on log2(fold change) <1 or <1 (respectively) and p-value<0.05". Please correct it.

Corrected (lines 1042-1044).

f) The authors wrote, "Protein abundance was quantitated based on the total ion count for all identified peptides in each sample, and normalized after log transform according to previously published protocols (60)". The authors should state in the methods section how the normalization was performed (rather than just referring to a paper).

Added (lines 1044-1053).

g) The tables containing proteomics data look better, my additional suggestion would be to:1) Add a unique peptide count per replicate (this parameter is more informative than peptide count).

Added for all datasets (Supplementary Files 1, 2).

2) Add a score parameter for protein identification.

We have added Sum PEP and Sequest HT scores for all datasets (Supplementary Files 1, 2).

5) Since the proteome coverage in the WCL samples was rather low (2.6-3k proteins, of which only 396-407 are annotated as mitochondrial in MitoCarta3.0), the authors were asked to analyze proteins in isolated mitochondrial fractions. The authors followed the recommendation only for ndufa9-/-, however, they did not do it for cox10-/-.The additional analysis they performed yielded quantification data for 738 mitochondrial proteins in isolated mitochondria compared to 396 mitochondrial proteins in WCL samples, which clearly shows an improvement. As the proteomic analysis of isolated mitochondria ndufa9-/- vs ndufa9f/f in fact nearly doubled the number of quantified mitochondrial proteins reported, the authors should also include a similar analysis for the isolated mitochondria cox10-/- vs cox10f/f.

Thank you for this suggestion. We have compiled an analogous dataset comparing proteomes from enriched mitochondrial fractions in cox10^-/-^ and cox10^f/f^ livers, which is presented in Supplementary File 2 and is being deposited in Pride. We detected 721 mitochondrial proteins, which was comparable to the enrichment observed in our ndufa9-/- dataset.

We have created an additional supplementary figure (Figure 5 —figure supplement 3) to show this data, and discuss within the text (lines 351-361). We note that several components of complex IV are downregulated in cox10-/- livers, consistent with destabilization of complex IV. Components of other ETC components are spared or slightly unregulated. We note that there are a number of other changes in the mitochondrial proteome of cox10-/- livers, including downregulation of some fatty acid related proteins, and upregulation of proteins related to ubiquinol homeostasis.

6) They were asked to perform GO term enrichment analysis for the cox10-/- vs cox10f/f WCL dataset. The authors performed the GO term enrichment analyses separately for mitochondrial and other than mitochondrial proteins, however, both mitochondrial and non-mitochondrial proteins may be associated with certain GO terms. The authors should add an additional term enrichment analysis for all the proteins upregulated or downregulated in the cox10-/- vs cox10f/f WCL dataset.

We apologize for the confusion. In the previous version of the manuscript, we did perform GO term enrichment analysis for all up or down-regulated proteins, as well as non-mitochondrial proteins separately. We now refer to these analyses as “all” and “non-mitochondrial” for clarity.

We now provide additional analyses of mitochondrial proteins only in Figure 5 —figure supplements 1B and 2B; these new analyses are referred to as “mitochondrial”, and are also listed in Supplementary File 2.

7) The authors were asked to provide a correlation between the observed changes in the proteome and metabolome in cox10-/- livers. Specifically, they were asked to examine whether changes in the levels of specific metabolites result from the up- or down-regulation of proteins of the respective pathways.In response, the authors added Figure 5 —figure supplement 4 and included a short description in the manuscript. However, no specific examples were presented, e.g. up- or down-regulated level of enzyme A possibly contributed to up- or down-regulated level of metabolite X. Rather than (or in addition to) Figure 5 —figure supplement 4, another figure or a table linking specific enzymatic activities with metabolites should be included.

As suggested, we have supplemented the previous analysis with a pathway specific analysis focusing on metabolic enzymes and the metabolites they produce/consume. These diagrams are provided now in Figure 5 —figure supplement 5,6,7. We have expanded our text to include a discussion of these results (lines 371-427).

To summarize briefly, we find that fatty acid synthesis and fatty acid oxidation enzymes are largely changing in the opposite directions as would be expected for an increase in acyl-carnitine species (Figure 5 —figure supplement 5), and we suspect that the buildup of acyl-carnitine species is instead due to a direct inhibition of fatty acid oxidation secondary to loss of complex IV. For bile acid species, the majority of direct bile-acid metabolizing enzymes were not detectable in our dataset (Figure 5 —figure supplement 6). For electron carriers, we did observe some interesting trends: While the precise reasons for decreases in electron carrier abundance is not clear at this stage, there were some instances were consuming enzymes were upregulated (e.g., for pyruvate, α-hydroxybutyrate), or synthesis / recycling enzymes were downregulated (e.g., for α-hydroxybutyrate, tetrahydrobiopterin).